# Bacterial cell cycle control by citrate synthase independent of enzymatic activity

Matthieu Bergé[1]*, Julian Pezzatti[2], Víctor González-Ruiz[2,3], Laurence Degeorges[1], Geneviève Mottet-Osman[1], Serge Rudaz[2,3], Patrick H Viollier[1]*

[1]Department of Microbiology and Molecular Medicine, Faculty of Medicine, University of Geneva, Geneva, Switzerland; [2]Institute of Pharmaceutical Sciences of Western Switzerland (ISPSO), University of Geneva, Geneva, Switzerland; [3]Swiss Centre for Applied Human Toxicology (SCAHT), Basel, Switzerland

**Abstract** Proliferating cells must coordinate central metabolism with the cell cycle. How central energy metabolism regulates bacterial cell cycle functions is not well understood. Our forward genetic selection unearthed the Krebs cycle enzyme citrate synthase (CitA) as a checkpoint regulator controlling the $G_1 \to S$ transition in the polarized alpha-proteobacterium *Caulobacter crescentus*, a model for cell cycle regulation and asymmetric cell division. We find that loss of CitA promotes the accumulation of active CtrA, an essential cell cycle transcriptional regulator that maintains cells in $G_1$-phase, provided that the (p)ppGpp alarmone is present. The enzymatic activity of CitA is dispensable for CtrA control, and functional citrate synthase paralogs cannot replace CitA in promoting S-phase entry. Our evidence suggests that CitA was appropriated specifically to function as a moonlighting enzyme to link central energy metabolism with S-phase entry. Control of the $G_1$-phase by a central metabolic enzyme may be a common mechanism of cellular regulation.

*For correspondence:
matthieu.berge@unige.ch (MBé);
patrick.viollier@unige.ch (PHV)

Competing interests: The authors declare that no competing interests exist.

## Introduction

Nutritional control of cellular development and cell cycle progression have been described in many systems, but molecular determinants that govern the responses are known in only a few instances. Bacteria are attractive models for the elucidation of the underlying mechanisms because of their genetic tractability, their apparent morphological and cellular simplicity, and the robust influence of changing nutritional states on their growth and morphology. Links between central metabolism and the bacterial cell-cycle have been described, and three cases are known in which proteins resembling metabolic enzymes execute an important regulatory step in the early stages of cell division (*Monahan and Harry, 2016*). Such metabolic enzymes, often enzyme paralogs, that are appropriated for regulatory functions instead of or in addition to their normal enzymatic functions have been called moonlighting or trigger enzymes. Their enzymatic ancestry makes them ideal coupling factors to coordinate regulatory changes in response to metabolic fluctuations (*Commichau and Stülke, 2015*; *Huberts and van der Klei, 2010*), for example in bacterial cell cycle control.

The synchronizable α-proteobacterium *Caulobacter crescentus* is the preeminent model for elucidating fundamental cell cycle control mechanisms (*Hallez et al., 2017*). Cell division in *C. crescentus* is asymmetric and thus yields two dissimilar daughter cells. One daughter cell is a stalked and capsulated S-phase cell that replicates its genome before dividing. The other is a piliated and flagellated dispersal (swarmer) cell that resides in the non-replicative and non-dividing $G_1$-phase (*Figure 1A*). The old pole of the stalked cell features a cylindrical extension of the cell envelope, whereas that of the swarmer cell is decorated with a single flagellum and several adhesive pili. The placement and construction of organelles at the correct cell pole is dictated by the prior

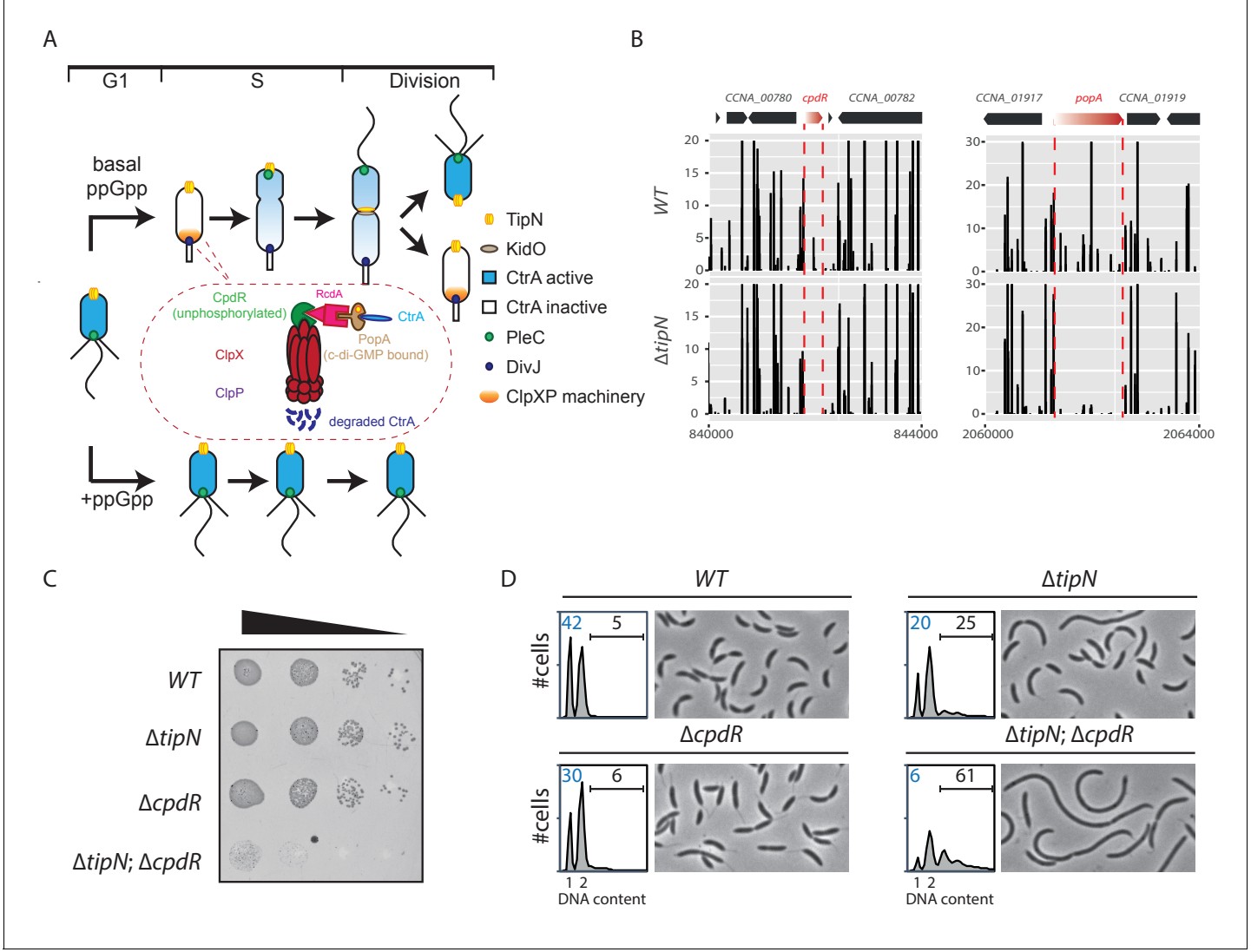

**Figure 1.** Synthetic sick interaction between *tipN* and proteolytic adaptor genes of the ClpXP machinery. (**A**) Schematic of the different stages of the *C. crescentus* cell cycle (G₁ phase, S phase and division are shown) in the normal condition (upper part). TipN (yellow dot) and KidO (brown circle) localization are represented throughout the cell cycle. Phosphorylated CtrA (blue) activates the transcription of G₁ phase genes and prevents DNA replication in the swarmer cell. Upon transition from a swarmer to stalked cell, the ClpXP machinery (orange) and its adaptors CpdR (green component in the encircled ClpXP machinery), RcdA (pink component) and PopA (brown component) localize to the incipient stalked pole where it degrades CtrA, allowing DNA replication and cell division. In the pre-divisional cell, the antagonistic kinase/phosphatase pair, DivJ (purple dot) and PleC (green dot) indirectly influence the phosphorylation of CtrA with the stalked cell compartment or swarmer cell compartment, respectively. PleC promotes CtrA phosphorylation in the swarmer cell whereas DivJ prevents its phosphorylation in the stalked cell. Pili and flagella are depicted as straight and wavy lines, respectively. In the case of ppGpp production occurring under conditions of carbon or nitrogen starvation, the swarmer to stalked cell transition is prevented (bottom part). (**B**) Transposon libraries were generated in the wildtype (*WT*) and the Δ*tipN* mutant (MB556). The sites of *Tn* insertion were identified by deep sequencing and mapped onto the *C. crescentus* NA1000 reference genome (nucleotide coordinates depicted on the X-axis). Two regions of the genome are depicted. The height of each line reflects a relative number in sequencing reads (Y-axis) at a given nucleotide position, and all the graphs for *WT* and Δ*tipN* are scaled similarly. *Tn* insertions in *cpdR* and *popA* were reduced in the Δ*tipN* mutant when compared to the *WT*. (**C**) EOP (efficiency of plating) assays showing spot dilutions of the indicated strains (MB1 [*WT*], MB556 [Δ*tipN*], MB2001 [Δ*cpdR*], and MB2017 [Δ*tipN*; Δ*cpdR*] from top to bottom). The four strains were grown overnight, adjusted at an OD₆₀₀ₙₘ of 0.5 and ten-fold serially diluted. Eight microliters of each dilution were spotted onto PYE plates. (**D**) Flow cytometry profiles and phase contrast images of *WT*, Δ*tipN*, Δ*cpdR* or *tipN*; Δ*cpdR* double mutants. Genome content (labelled as DNA content) was analyzed by fluorescence-activated cell sorting (FACS) during the exponential phase in peptone-yeast extract (PYE).

The online version of this article includes the following source data and figure supplement(s) for figure 1:

**Source data 1.** Tn-Seq data ratio comparing Tn insertion efficiency in *C. crescentus WT* and mutant strains measured as sequencing reads.
**Figure supplement 1.** Genetic interactions between *tipN* and the genes encoding proteoylctic adaptors.

*Figure 1 continued on next page*

*Figure 1 continued*

**Figure supplement 2.** Cell length control by KidO in cells lacking both TipN and CpdR.

recruitment of polar scaffolding proteins, including the TipN and PodJ coiled-coil proteins (*Figure 1A*; *Hinz et al., 2003*; *Huitema et al., 2006*; *Lam et al., 2006*; *Viollier et al., 2002*) and the PopZ polar organizer (*Bowman et al., 2008*; *Ebersbach et al., 2008*). As polar remodeling occurs as function of the cell cycle, it is not surprising that polarity determinants also affect progression of the cell division cycle (reviewed inby *Bergé and Viollier, 2018*).

The swarmer cell is obliged to differentiate into a stalked cell in order to complete the cell cycle. During the swarmer-to-stalked cell transition (also known as the $G_1 \rightarrow S$ transition), the flagellum is shed, pili are retracted, and a stalk is elaborated from the vacated pole while DNA replication competence is acquired (*Goley et al., 2007*; *Laub et al., 2007*). A critical regulatory protein that coordinates morphological and cell cycle stages is the essential <u>c</u>ell cycle <u>t</u>ranscriptional <u>r</u>egulator <u>A</u> (CtrA), a DNA-binding (OmpR-like) response regulator that, upon phosphorylation, directly binds and regulates the origin of replication (*ori*) (*Laub et al., 2000*; *Quon et al., 1996*; *Quon et al., 1998*) and the promoter regions of developmental genes, including those that are activated in $G_1$-phase (*Fiebig et al., 2014*; *Fumeaux et al., 2014*). CtrA activates the $G_1$-phase promoters of *pilA*, which encodes the structural subunit of the pilus filament (*Skerker and Shapiro, 2000*), several flagellin genes and other genes controlling cell envelope modification that are reviewed in *Ardissone and Viollier (2015)*.

CtrA is regulated at the level of activity by phosphorylation and at the level of stability by cell cycle-controlled proteolysis (*Figure 1A*), both controlled by a complex phospho-signaling pathway via the CckA histidine kinase/phosphatase (*Biondi et al., 2006*; *Domian et al., 1997*; *Jacobs et al., 1999*; *Tsokos et al., 2011*; *Wu et al., 1998*). A reversal of the CckA phosphoflux during the $G_1 \rightarrow S$ transition activates the branch controlling the degradation of CtrA (*Chen et al., 2009*; *Joshi and Chien, 2016*). This proteolytic pathway involves the protease ClpXP primed by three selectivity factors that present CtrA to ClpXP (*Figure 1A*). These proteolytic adaptors, CpdR, RcdA and PopA, are organized into a regulatory hierarchy that coordinates the degradation of multiple cell cycle-regulated proteins during the $G_1 \rightarrow S$ transition (*Duerig et al., 2009*; *Iniesta et al., 2006*; *Joshi et al., 2015*; *McGrath et al., 2006*). Upon degradation of CtrA, the DNA replication block is relieved and $G_1$-phase genes are no longer expressed. Thus, the maintenance of cells in the $G_1$ phase requires CtrA to remain present and phosphorylated (*Domian et al., 1997*; *Hung and Shapiro, 2002*).

Interestingly, the duration of the $G_1$ period is affected by nutrient availability in *C. crescentus* and other α-proteobacteria through a pathway involving CtrA (*De Nisco et al., 2014*; *Hallez et al., 2017*). Upon nitrogen or carbon starvation, the $G_1 \rightarrow S$ transition is blocked (*Britos et al., 2011*; *England et al., 2010*; *Gorbatyuk and Marczynski, 2005*; *Lesley and Shapiro, 2008*; *Leslie et al., 2015*). This $G_1$ block is associated with the accumulation of the guanosine tetra- and penta-phosphate [(p)ppGpp] alarmone (*Figure 1A*; *Boutte et al., 2012*; *Lesley and Shapiro, 2008*; *Ronneau et al., 2016*), which affects important cellular processes in bacteria such as transcription, translation or DNA replication (*Liu et al., 2015*; *Wang et al., 2019*; *Zhang et al., 2018*). Rsh family proteins directly modulate the intracellular level of (p)ppGpp and most bacterial genomes encode at least one bifunctional Rsh protein that is able to synthesize and hydrolyze (p)ppGpp. *C. crescentus* encodes a single bifunctional Rsh enzyme, named SpoT that produces (p)ppGpp in response to nutrient deprivation (*Atkinson et al., 2011*; *Boutte et al., 2012*; *Lesley and Shapiro, 2008*; *Ronneau et al., 2016*). Previous studies have shown that (p)ppGpp accumulation leads to a stabilization of CtrA by an unknown mechanism that impairs the $G_1 \rightarrow S$ transition (*Lesley and Shapiro, 2008*; *Leslie et al., 2015*). (p)ppGpp is required for efficient recruitment of CtrA to target promoters and for CtrA-dependent promoter activity in stationary phase cells, and this requirement can be suppressed by mutations in RNA polymerase (*Delaby et al., 2019*).

Here, we report that citrate synthase (CitA), the first enzyme of the Krebs (tricarboxylic acid [TCA]) cycle that catalyzes the reaction between oxaloacetate and acetyl-CoA to form citrate, fulfills an unprecedent role as a checkpoint regulator that controls the $G_1 \rightarrow S$ transition by acting negatively on CtrA. We show that loss of CitA leads to an accumulation of active CtrA, prolonging the $G_1$ phase provided (p)ppGpp is present. Although CitA is a functional citrate synthase, loss of

CitA does not lead to an insufficiency in energy and biosynthetic precursors, because the functional paralog CitB supports biosynthetic activity. Surprisingly, catalytically inactive CitA still retains cell cycle control functions, indicating that CitA acts as a moonlighting enzyme of central energy metabolism to regulate S-phase entry.

## Results

### $G_1$-phase defect in cells lacking TipN and adaptors of the ClpXP machinery

As the mild cell cycle defect of cells lacking the TipN polarity factor ($\Delta tipN$) is not well understood, we sought Tn mutations that enhance the defect. To this end, we compared the Tn insertion sequencing (Tn-Seq) profiles of wild-type and $\Delta tipN$ cells, seeking Tn insertions that specifically undermine the viability or fitness of cells lacking TipN. This analysis revealed Tn insertions in *cpdR*, *rcdA* or *popA* genes that are underrepresented in $\Delta tipN$ cells when compared with *WT* cells (*Figure 1B* and *Figure 1—figure supplement 1A*, *Figure 1—source data 1*). These three genes encode a hierarchical proteolytic adaptor cascade that coordinates the delivery of a range of substrates, including CtrA, to the ClpXP protease for proteolytic removal during the $G_1 \rightarrow S$ transition (*Duerig et al., 2009*; *Iniesta et al., 2006*; *Joshi et al., 2015*; *Joshi and Chien, 2016*; *McGrath et al., 2006*). A converse Tn-Seq comparison between *WT* and $\Delta cpdR$ cells also revealed an underrepresentation of Tn insertions in the *tipN* gene (*Figure 1—figure supplement 1B*, *Figure 1—source data 1*). To confirm the genetic relationship between *tipN* and *cpdR, rcdA* or *popA*, we created double mutants by introducing the $\Delta cpdR$, $\Delta rcdA$ or $\Delta popA$ mutations into $\Delta tipN$ cells and found that all of the resulting double mutants exhibit a reduction in viability by three orders of magnitude on a logarithmic scale, as determined by efficiency of plating (EOP) assays (*Figure 1C*; *Figure 1—figure supplement 1C and D*).

Examination of $\Delta tipN$; $\Delta cpdR$ double mutant cells by phase-contrast microscopy showed that they are 70% more elongated on average than *WT* and $\Delta cpdR$, $\Delta rcdA$ or $\Delta popA$ single mutant cells (*Figure 1D* and *Figure 1—figure supplement 1E and F*). Flow cytometry analysis of exponentially growing $\Delta tipN$; $\Delta cpdR$ double mutant cells using a fluorescence activated cell sorter (FACS) revealed a massive reduction in the number of $G_1$-phase cells and an increase in the frequency of cells with multiple (>2) chromosomes compared to *WT*, whereas $\Delta cpdR$ and $\Delta tipN$ single mutants only showed a slight decrease in the $G_1$ population (*Figure 1D*). Importantly, the $\Delta tipN$; $\Delta rcdA$ and $\Delta tipN$; $\Delta popA$ double mutants show a similar accumulation of elongated cells and reduction in the number of $G_1$-phase cells (*Figure 1—figure supplement 1E and F*). Thus, the proteolytic adaptors promote efficient cell cycle progression in cells that lack TipN.

### Indirect effect of proteolytic adaptors on CtrA in the $\Delta tipN$ mutant

A reduction in the proportion of $G_1$ cells is often correlated with reduced activity or abundance of CtrA, the principal $G_1$-phase transcriptional regulator. To assess whether this is also the case in cells that lack both TipN and CpdR, we introduced a translational *pilA*::$P_{pilA}$-*GFP* promoter probe reporter into the *pilA* locus of *WT* cells, $\Delta tipN$ and $\Delta cpdR$ single mutant cells, and $\Delta tipN$; $\Delta cpdR$ double mutant cells. In this reporter, the CtrA-dependent *pilA* promoter ($P_{pilA}$) that fires in $G_1$-phase along with the PilA start codon is translationally fused to a start codon-less variant encoding the green fluorescent protein (GFP). GFP expression from this reporter can be conveniently observed and quantified by live-cell fluorescence microscopy (*Figure 2A*). In agreement with the FACS profiles shown in *Figure 1E*, GFP fluorescence intensity is only slightly lower in $\Delta cpdR$ cells than in *WT* cells, but clearly reduced in $\Delta tipN$ cells. Importantly, a further strong decrease in GFP fluorescence is observed in $\Delta tipN$; $\Delta cpdR$ double mutant cells, indicating a strong downregulation in CtrA-dependent reporter activity. Likewise, transcription from a reporter in which $P_{pilA}$ is fused to the promoterless *nptII* gene (conferring resistance to kanamycin) at the *pilA* locus (*pilA*::$P_{pilA}$-*nptII*) is strongly reduced in $\Delta tipN$; $\Delta cpdR$ double mutant cells when compared to *WT* cells, precluding growth on plates containing 20 µg/mL kanamycin (*Figure 2B*). We conclude that cells that lack both TipN and CpdR suffer from an insufficiency of CtrA.

Paradoxically, inactivation of CpdR, RcdA or PopA should enhance CtrA abundance in $\Delta tipN$ cells, since the proteolytic removal of CtrA at the $G_1 \rightarrow S$ transition should be blocked in the absence of

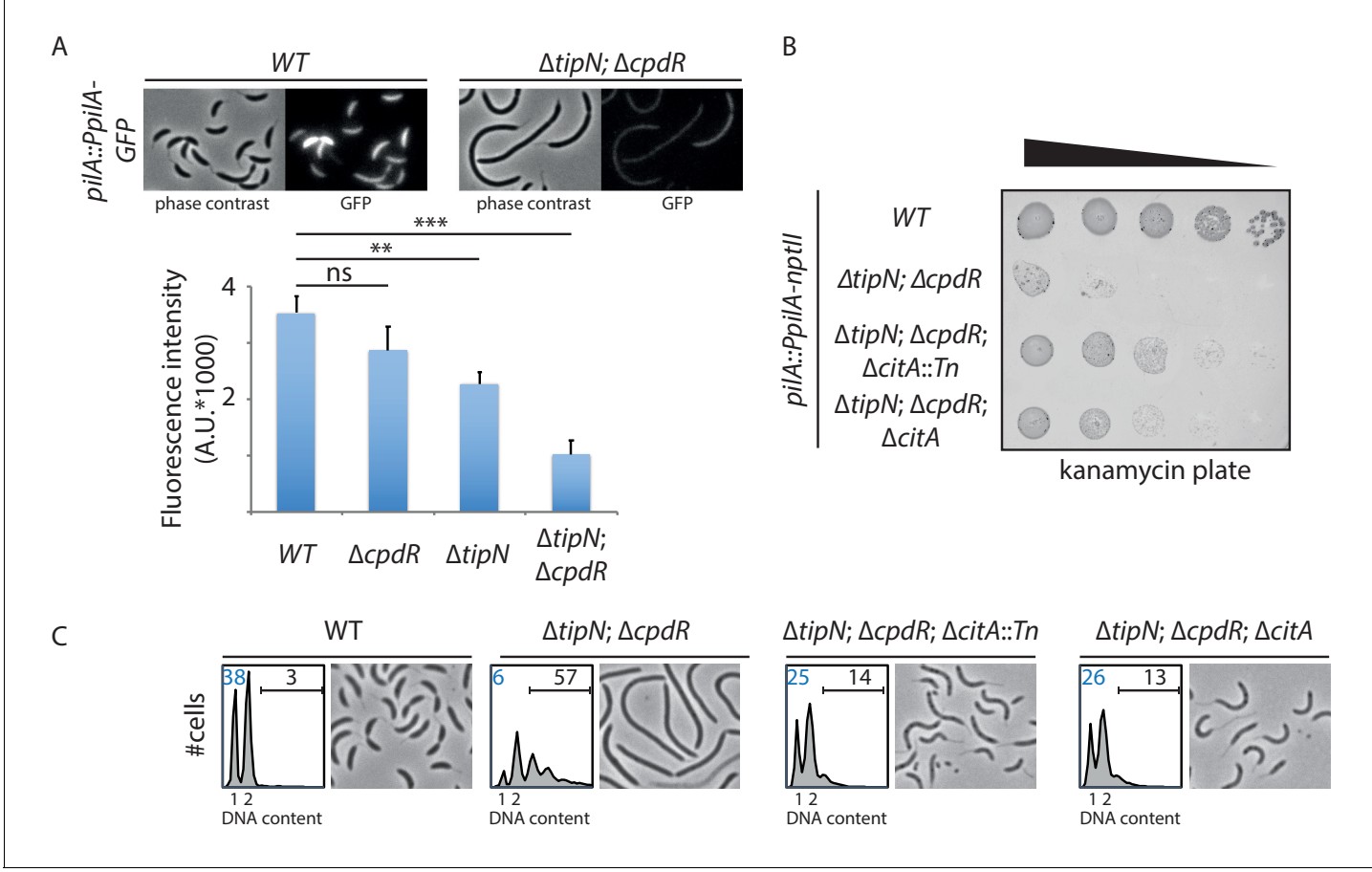

**Figure 2.** Genetic screen to identify Tn insertions that enhance CtrA. (**A**) CtrA activity in *WT* (MB2325), *ΔtipN* (MB2337) and *ΔcpdR* (MB2329) single mutant cells, and in *ΔtipN ΔcpdR* (MB2331) double mutant cells, was monitored using a *pilA::PpilA-GFP* transcriptional reporter whose activity is dependent on CtrA. Fluorescence intensity was automatically quantified, and t-tests were performed to determine the significance with p<0.05 (**) and p<0.005 (***). (**B**) Spot dilutions of the indicated strains (MB2268 [*WT*], MB2271 [*ΔtipN*; *ΔcpdR*], MB3056 [*ΔtipN*; *ΔcpdR*; *ΔcitA*::*Tn*], and MB3058 [*ΔtipN*; *ΔcpdR*; *ΔcitA*] from top to bottom) carrying the *pilA*::P$_{pilA}$-*nptII* transcriptional reporter on PYE plates containing kanamycin (20 µg.mL$^{-1}$). (**C**) FACS profiles and phase contrast images of the strains shown in panel (B). FACS profiles showing genome content (ploidy) of cells growing exponentially in PYE and then treated with rifampicin (20 µg.mL$^{-1}$) for 3 hours to inhibit DNA replication. Numbers (%) of G$_1$-phase cells and cells containing more than two chromosomes are indicated in blue and black, respectively.

The online version of this article includes the following figure supplement(s) for figure 2:

**Figure supplement 1.** Loss of CitA attenuates the cell division defect of *tipN cpdR* double mutant cells.

each of these adaptors. We therefore reasoned that another, indirect, effect underlies the crippled CtrA activity in *ΔtipN* ; *ΔcpdR* cells. It could be that this effect is mediated by an inhibitor of the CtrA pathway that is also degraded by the CpdR-RcdA-PopA pathway, which would accumulate in the absence of CpdR. We sought to uncover this gene by comparative Tn-Seq analyses in *ΔtipN*; *ΔcpdR* double mutant cells *versus WT* or *ΔtipN* and *ΔcpdR* single mutant cells, and we found a 19-fold increase in Tn insertions in the *kidO* gene (***Figure 1—figure supplement 2A***, ***Figure 1—source data 1***). KidO is a bifunctional oxidoreductase-like negative regulator of cell division and the CtrA pathway (***Radhakrishnan et al., 2010***). Akin to CtrA, KidO is degraded during the G$_1$→S transition by (CpdR/RcdA/PopA) adaptor-primed ClpXP. As KidO is stabilized in cells lacking CpdR, RcdA or PopA (***Radhakrishnan et al., 2010***), we asked whether the stabilization of KidO confers the cell defect of *ΔtipN*; *ΔcpdR* cells. To test this idea, we expressed the *kidO*$^{AA::DD}$ allele from the *xylX* locus in *ΔtipN* cells. This allele encodes a mutant form of KidO in which the two penultimate alanine residues are both substituted by aspartic acid residues, a double substitution that prevents degradation of KidO by the ClpXP protease at the G$_1$→S transition, akin to the *ΔcpdR* mutation (***Radhakrishnan et al., 2010***). The resulting *ΔtipN xylX::kidO*$^{AA::DD}$ cells are highly filamentous, even

without induction of the *xylX* promoter by xylose: with more than two chromosomes recapitulating the phenotype of the Δ*tipN*; Δ*cpdR* double mutant cells (*Figure 1—figure supplement 2B*). Conversely, an in-frame deletion in *kidO* (Δ*kidO*) restores a near *WT* cell division phenotype to Δ*tipN*; Δ*cpdR* cells (*Figure 1—figure supplement 2B*).

Taken together, these experiments support the conclusion that stabilization of KidO strongly impairs progression of the cell division cycle in cells lacking TipN.

## Genetic screen to identify regulators of the G$_1$ to S transition

The impaired activity of the *pilA*::P$_{pilA}$-*nptII* reporter in Δ*tipN*; Δ*cpdR* cells (*Figure 2A*) offered a convenient opportunity to isolate Tn insertions that restore or elevate P$_{pilA}$ activity. Towards this goal, we mutagenized Δ*tipN*; Δ*cpdR*; *pilA*::P$_{pilA}$-*nptII* reporter cells with a mini-*himar1* Tn (Mar2xT7) encoding gentamycin resistance, and selected for growth on plates containing kanamycin and gentamycin. Among several isolated mutants, we found one mutant harboring a Tn insertion in the middle of the *CCNA_01983* (henceforth *citA*) gene, whose gene product is annotated as a type II citrate synthase (PRK05614). After confirming by backcrossing that the *citA*::Tn mutation did indeed confer kanamycin resistance to Δ*tipN*; Δ*cpdR*; *pilA*::P$_{pilA}$-*nptII* reporter cells, we engineered an in-frame deletion of *citA* (Δ*citA*) and found that this mutation also supports growth of Δ*tipN*; Δ*cpdR*; *pilA*::P$_{pilA}$-*nptII* reporter cells on kanamycin plates, indicating that inactivation of *citA* augments P$_{pilA}$ activity (*Figure 2B*). Moreover, the *citA*::Tn or the Δ*citA* mutations both correct the abnormal cell size distribution (cell filamentation) and augment the G$_1$ population of Δ*tipN*; Δ*cpdR* double mutant cells (*Figure 2C* and *Figure 2—figure supplement 1A*).

In sum, inactivation of *citA* gene causes a strong increase of P$_{pilA}$ activity and promotes the accumulation of G$_1$ cells in the joint absence of TipN and CpdR.

## CitA encodes a citrate synthase

The primary structure of CitA resembles that of citrate synthases, which execute the first enzymatic reaction in the Krebs (tricarboxylic, TCA) cycle in which the acetyl group from acetyl-CoA is condensed onto oxaloacetate to form citrate (*Figure 3—figure supplement 1A*; *Figure 3A*). *C. crescentus* CitA has 65% amino acid identity to the GltA citrate synthase from *Escherichia coli* K12 (strain MG1655) and 32% identity to CitA from *Bacillus subtilis* (strain 168). To confirm that *C. crescentus* CitA does indeed have citrate synthase activity, we probed for heterologous complementation of glutamate auxotrophy in *E. coli* Δ*gltA* cells that lack citrate synthase activity (*Lakshmi and Helling, 1976*). To this end, we engineered *E. coli* Δ*gltA* cells expressing either *C. crescentus* CitA or *E. coli* GltA from a multicopy plasmid. As expected, *E. coli* Δ*gltA* cells harboring the empty vector were unable to grow in (M9) minimal medium without glutamate, but Δ*gltA* cells grew well in the presence of either the *gltA*- or the *citA*-expression plasmid (*Figure 3B*). Thus, *C. crescentus citA* encodes a functional citrate synthase.

Next, we conducted metabolic profiling experiments using liquid chromatography coupled to high-resolution mass spectrometry (LC-HRMS) to quantify the abundance of intracellular metabolites in *C. crescentus WT* and *citA*::Tn or Δ*citA* cells grown in PYE (*Pezzatti et al., 2019a*). Robust quantitation of 103 metabolites (*Figure 3—source data 1*) revealed that the metabolomic profile of *citA*::Tn resembles that of Δ*citA* cells. Surprisingly, these metabolomic analyses did not show any significant difference in many TCAs such as citrate and isocitrate when comparing *WT* and *citA* mutant cells (*Figure 3—figure supplement 1B*). An indication that TCA cycle flux is nevertheless affected in the absence of CitA comes from the observation that there is a small increase in the levels of acetyl-CoA, as would be expected for citrate synthase mutant cells (*Figure 3C*).

The relatively modest effect of the Δ*citA* mutation on the TCA cycle activity might result from the presence of a protein(s) other than CitA that has citrate synthase activity. Unlike other TCA cycle enzymes, CitA is not essential for the viability of *C. crescentus* cells on PYE (*Christen et al., 2011*). Therefore, we reasoned that CitA is not the only citrate-synthase-like protein encoded in the *C. crescentus* genome. Indeed, BLAST searches revealed the presence of two other putative citrate synthase genes: *CCNA_03757* and *CCNA_03758* (*Figure 3—figure supplement 1A*) (henceforth *citB* and *citC*, respectively), which were also annotated as non-essential for viability on PYE (*Christen et al., 2011*). The *citB* and *citC* genes encode proteins with 30% and 32% identity to *CitA* from *C. crescentus*, 30% and 33% identity to *GltA* from *E. coli* K12 (MG1655), and 37% and 32%

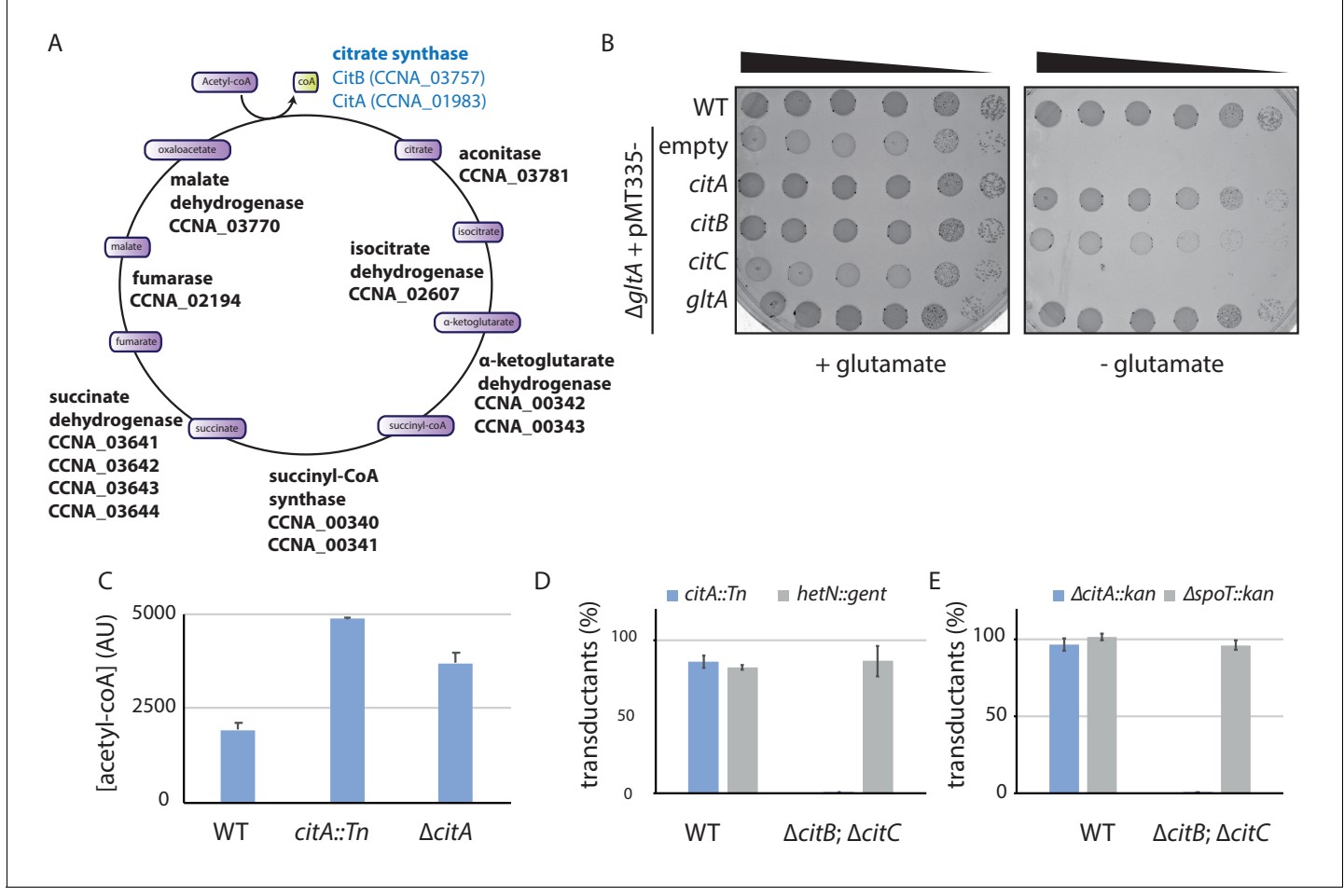

**Figure 3.** The *C. crescentus* genome encodes two functional citrate synthases. (**A**) A schematic of the Krebs cycle and the corresponding gene products in *C. crescentus*. The two functional citrate synthases are indicated in blue. Essential gene products, as inferred from Tn-Seq (*Christen et al., 2011*), are highlighted in bold. (**B**) Spot dilutions (EOP assays) of the indicated *WT* and Δ*gltA E. coli* strains (eMB554 [*WT*], eMB556 [Δ*gltA* + empty], eMB558 [Δ*gltA* + *citA*], eMB560 [Δ*gltA* + *citB*], eMB562 [Δ*gltA* + *citC*] and eMB564 [Δ*gltA* + *gltA* ] from top to bottom) on minimal medium containing glutamate or not. Only the strain carrying a functional citrate synthase can grow without glutamate. (**C**) LC-MS-based quantification of acetyl-CoA in extract of *WT* (MB1), *citA*::Tn (MB2622) and Δ*citA* (MB2559) cells grown in PYE liquid cultures. Error bars denote the standard deviation of the mean from three biological replicates. (**D**) ΦCR30-mediated generalized transduction frequencies of *citA*::Tn into *WT* (MB1) or Δ*citBC* double mutant cells (MB2679). For transduction, cells were normalized according to $OD_{600nm}$ ~1 and infected with the same amount of phage lysates from *citA*::Tn cells or with phage lysates from cells with a transposon insertion in the *hetN* gene (encoding gentamycin resistance) as a control for transduction. The transductants were selected on PYE plates containing gentamycin. The numbers of transduced colonies were counted after 3 days of incubation at 30° C. Error bars denote the standard deviation of the mean for three independent experiments. Cells harboring the Δ*citBC* mutation are not able to accept the *citA*::Tn mutation. (**E**) Same as in panel (**D**) using the Δ*citA*::*kan* allele or a deletion in the *spoT* gene (encoding kanamycin resistance, Δ*spoT*::*kan*) delivered by ΦCR30-mediated generalized transduction. Transductants were selected on PYE plates containing kanamycin.

The online version of this article includes the following source data and figure supplement(s) for figure 3:

**Source data 1.** Metabolomic data sets showing the metabolites detected (sheet 1) and statistically significant changes in relative metabolite abundance between *WT* and *citA*::Tn mutant cells (sheet 2) or between *WT* and Δ*citA* mutant cells (sheet 3), presented as volcano plots.

**Figure supplement 1.** Primary structure alignment of CitA and homologs.

identity to *CitA* from *Bacillus subtilis* strain 168. We therefore tested the ability of *citB* and *citC* to support citrate synthase function by heterologous complementation of the glutamate auxotrophy of *E. coli* Δ*gltA* cells on minimal medium lacking glutamate, and found that expression of CitB, but not CitC, supported growth (*Figure 3B*). Thus, *C. crescentus citB* also encodes a functional citrate synthase and *citA* mutants are probably able to grow on PYE because of residual citrate synthase activity conferred by CitB. To test whether CitA is essential in cells lacking both *citB* and *citC*, we first created a strain with in-frame deletions in *citB* and *citC* (Δ*citBC*) and then attempted to introduce

*citA*::Tn (which encodes gentamycin resistance) or Δ*citA* (tagged with a kanamycin resistance marker, Δ*citA*::pNPTS138) by φCr30-mediated generalized transduction. Unlike *WT* cells, Δ*citBC* cells do not accept *citA*::Tn or Δ*citA*::pNPTS138 generalized transducing particles (*Figure 3D*), but accept generalized transducing particles harboring another genomic locus marked with either the gentamycin or the kanamycin resistance gene with efficiency similar to that of *WT* cells. We conclude that *C. crescentus* encodes at least two functional citrate synthases, one of which is absolutely required for growth on PYE.

## CitA promotes S-phase entry, independently of its enzymatic activity

To determine how loss of CitA signals $G_1$ cell accumulation, we combined population-based and single-cell approaches. First, EOP assays and growth curve measurements indicate that the absence of CitA leads to a slow growth phenotype in PYE and that CitA is required for growth on minimal M2G medium (*Figure 4A*). Phase contrast microscopy of *citA*::Tn or Δ*citA* mutant cells revealed that Δ*citA* cells are shorter and narrower than *WT* cells (areas of 0.42 ± 0.009 μm and 0.43 ± 0.007 μm, respectively, for the *citA*::Tn and Δ*citA* compared to 0.69 ± 0.01 μm for *WT* cells; *Figure 4B*), perhaps because they spend more time in the non-growing $G_1$ phase. Indeed, FACS profiles revealed a strong increase in the $G_1$-phase population in the absence of CitA: 68.3 ± 1.25% and 69.3 ± 1.22 of *citA*::Tn and Δ*citA* cells, respectively, reside in $G_1$ phase compared to 36.1 ± 0.6% of *WT* cells (*Figure 4B*). Importantly, these phenotypes of *citA* mutant cells cannot be corrected by the addition of exogenous glutamine and, therefore, are not related to glutamine auxotrophy. Indeed, the addition of glutamine to PYE or to M2G (minimal medium) does not ameliorate growth or division, as determined by EOP assays (*Figure 4—figure supplement 1A*). Moreover, the addition of glutamine does not restore a normal FACS profile to *citA* mutant cells (*Figure 4—figure supplement 1B*). The *citA* mutant phenotypes are not corrected by complementation of *citA* mutant cells with a multi-copy plasmid harboring *C. crescentus citB* (pMT335-*citB*) or *E. coli gltA* (pMT335-*gltA*), arguing that these functions probably depend on the presence of the CitA protein rather than on citrate synthase enzymatic activity (*Figure 4C*). However, these deficiencies are corrected when a *WT* copy of *citA* is expressed *in trans* on a multi-copy plasmid (pMT335-*citA*) (*Figure 4C*). Thus, CitA promotes the $G_1$→S transition, a function that other citrate synthases such as CitB and GltA cannot provide.

Further support for the conclusion that CitA fulfills a regulatory role that is independent of its catalytic activity came from the discovery that catalytically inactive CitA can still control the cell cycle. Residue H306 of *E. coli* GltA is critical to bind oxaloacetate, and its substitution impairs the catalytic activity of GltA (*Handford et al., 1988*; *Pereira et al., 1994*). We thus engineered variants in which the corresponding residue (H303) in *C. crescentus* CitA is substituted either by a tryptophan or by an alanine, giving rise to the H303W and H303A CitA variants. As expected, expression of the CitA[H303W] or CitA[H303A] variant in *E. coli* Δ*gltA* cells no longer correct the glutamate auxotrophy on minimal medium, as determined by EOP assays (*Figure 4—figure supplement 1C*). Immunoblotting using polyclonal antibodies to CitA revealed that these variants are produced to the same levels as WT CitA (*Figure 4—figure supplement 1D*). We therefore conclude that CitA[H303W] and CitA[H303A] have lost enzymatic activity. When these variants are expressed in *C. crescentus* Δ*citA* mutant cells to similar levels as WT CitA (*Figure 4—figure supplement 1E*), a normal FACS profile and cell size distribution is observed by phase-contrast microscopy (*Figure 4D*). As these results show that the catalytic activity of CitA is dispensable for its developmental function, CitA must fulfill a specific regulatory role in promoting the $G_1$→S transition.

To establish that CitA is required for the $G_1$→S transition, we performed cell cycle studies using synchronized *WT* and *citA* mutant cells. FACS profiles revealed that *WT* $G_1$ cells initiate DNA replication 30 minutes after their release into PYE, whereas *citA*::Tn or Δ*citA* $G_1$ cells do not enter S-phase before 90 minutes after their release into PYE (*Figure 4—figure supplement 1G*). We also discovered that a fraction of *citA*::Tn or Δ*citA* cells remain in $G_1$ phase, with only approximately half entering S-phase. To confirm this observation at the single-cell level, we conducted time-lapse microscopy experiments with synchronized *WT* and *citA*::Tn or Δ*citA* G1 cells expressing GFP-ParB as a marker for DNA replication (*Figure 4E*). ParB is a chromosome partitioning protein that specifically binds near the origin of replication ($C_{ori}$) and is translocated with a duplicated copy of $C_{ori}$ to the daughter cell pole once DNA replication commences (*Mohl and Gober, 1997*; *Thanbichler and Shapiro, 2008*). In synchronized *WT* G1 cells expressing ParB-GFP, we observed a single, polarly

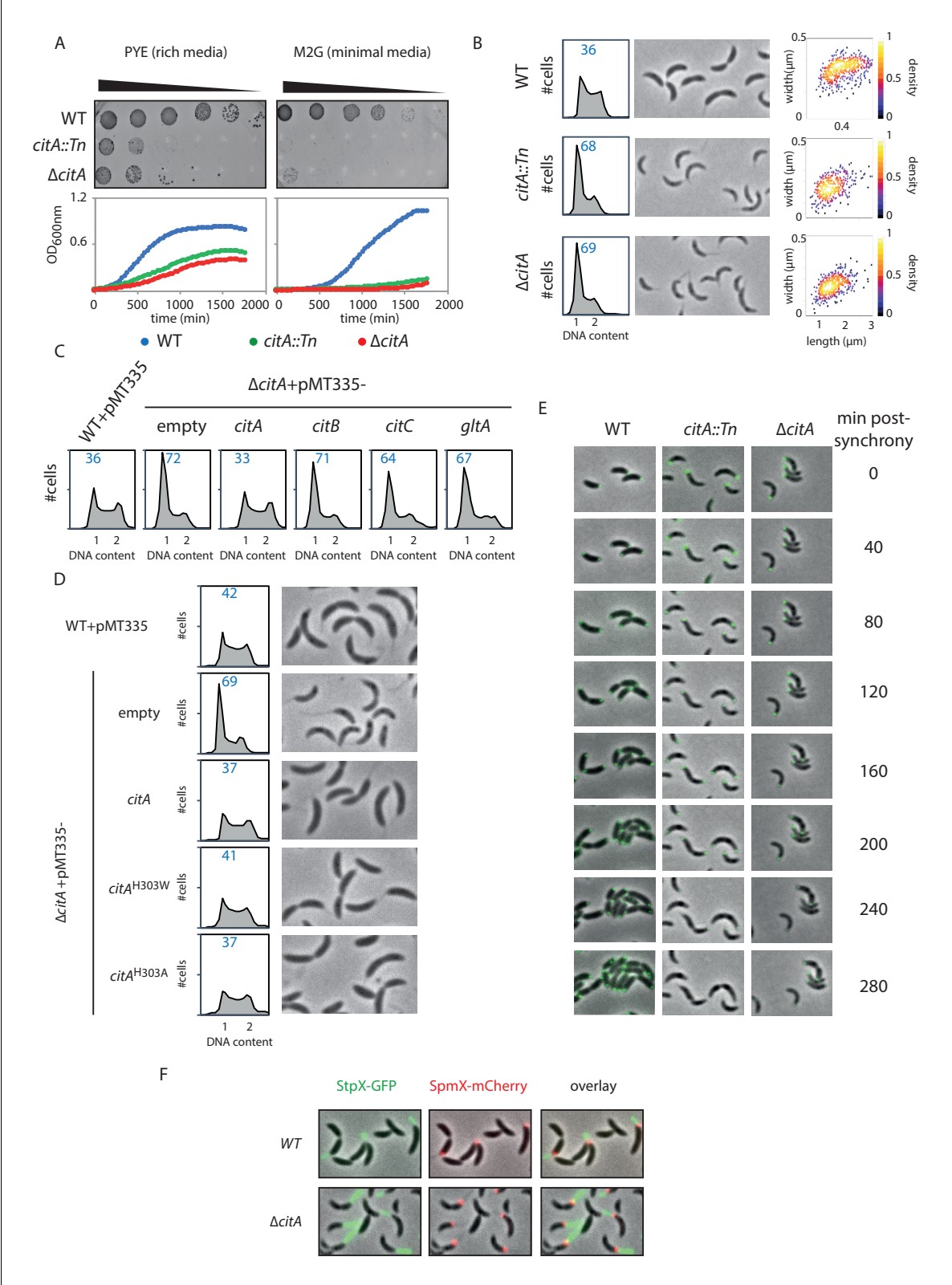

**Figure 4.** Inactivation of CitA induces a G₁ block. (**A**) Spot dilution (EOP assays) and growth curve measurements of *WT* (MB1), *citA::Tn* (MB2622) and Δ*citA* (MB2559) cells. For spot dilution, cells were grown overnight in PYE, adjusted to OD$_{600nm}$ ~0.5, and serially diluted on a rich (PYE) medium (left upper part) or on a minimal (M$_2$G) medium (right upper part). For the growth curves, cells were grown overnight in PYE, washed twice with M$_2$ buffer, and a similar amount of cellswas used to inoculate PYE medium (left bottom part) or M$_2$G medium (right bottom part). (**B**) FACS profiles and phase

*Figure 4 continued on next page*

*Figure 4 continued*

contrast images of *WT* (MB1), *citA*::Tn (MB2622) and Δ*citA* (MB2559) cells growing exponentially in PYE. The right panel shows a scatter plot of the cell lengths and widths of each indicated population. (**C**) FACS profiles of Δ*citA* cells harboring an empty plasmid (MB3433), or a derivative with *citA* (MB3435), *citB* (MB3469), or *citC* (MB3471) from *C. crescentus,* or the citrate synthase gene (*gltA*) from *E. coli* (MB3473). *WT* cells harboring an empty pMT335 are also shown (MB1537). (**D**) FACS profiles and phase-contrast images of *C. crescentus* expressing a catalytic mutant of CitA. *WT* cells carrying an empty plasmid (MB1537), or Δ*citA* cells harboring an empty plasmid (MB3433) or a derivative with *citA* (MB3435), *citA*[H303A] (MB3439) or *citA*[H303W] (MB3437) are shown. (**E**) Time-lapse fluorescence microscopy of *WT* (MB557), *citA*::Tn (MB2452) and Δ*citA* (MB3467) cells harboring a *parB::gfp-parB* reporter. Cells were grown in PYE, synchronized and spotted on a PYE agarose pad. A picture was taken every 20 minutes. (**F**) Fluorescence microscopy of *WT* (MB334) and *citA*::Tn (MB3598) cells harboring a *spmX::spmX-mCherry* or a *stpX::stpX-gfp* reporter. Cells were grown exponentially in PYE. Each fluorescence channel is shown alone or together superimposed on phase contrast images.

The online version of this article includes the following figure supplement(s) for figure 4:

**Figure supplement 1.** Addition of glutamine does not ameliorate the *citA* mutant phenotype.

localized $C_{ori}$, represented by a single GFP-ParB focus. After 40 minutes, ~80% (n = 39) of the cells have a duplicated GFP-ParB focus, one of which is segregated to the opposite pole. Finally, cell division is completed by 120 minutes. By contrast, in *citA*::Tn (n = 35) or Δ*citA* (n = 29) $G_1$ cells, a duplicated GFP-ParB focus only appeared in some cells after 100 minutes. Importantly, we noticed that even after 260 minutes, ~60% of the population still exhibit only one GFP-ParB focus. Thus, a large fraction of the population remains in $G_1$-phase and only part of the *citA* mutant population enters S-phase.

While chromosome duplication is delayed, *citA* $G_1$ cells harbor a long stalk, as indicated by live-cell fluorescence imaging with the stalk marker StpX-GFP (*Hughes et al., 2010*), which coincides with the presence of a SpmX-mCherry focus, a marker of the stalked pole (*Figure 4F*). Knowing that SpmX is normally absent from the $G_1$ cells, we hypothesized that polar remodeling and chromosome replication might be uncoupled in the absence of CitA (*Figure 5—figure supplement 1C*). Snapshot analyses of a population of synchronized *WT* cells expressing MipZ-YFP (a marker of chromosome origin) and SpmX-mCherry (a marker of polar remodeling) revealed that chromosome duplication occurs before the appearance of SpmX-mCherry. Contrary to that, 60 minutes after synchronization, *citA*::Tn cells have a MipZ-YFP and SpmX-mCherry focus at the same pole, suggesting that chromosome replication and polar remodeling is uncoupled in the absence of CitA. This is typically a phenotype observed in cells that have hyperactivation of CtrA (*Hung and Shapiro, 2002*).

## Loss of CitA enhances the abundance of active CtrA

As inactivation of *citA* augments $P_{pilA}$ activity, we hypothesized that the *citA* mutation elevates CtrA activity and/or abundance. To test this hypothesis, we asked whether the activity of other CtrA-activated promoters is also elevated in Δ*citA* cells compared to *WT* cells. LacZ-based promoter probe assays indeed revealed elevated activity of CtrA-dependent promoters (*Figure 5A*). To explore whether loss of CitA alters the levels of active CtrA, we used immunoblotting of Phos-tag PAGE to confirm that the levels of phosphorylated CtrA (CtrA ~P) in extracts of Δ*citA* cells are elevated relative to those in *WT* cell extracts (*Figure 5B and C*). This result prompted us to investigate whether this increase of CtrA steady-state levels was caused by increased stability of CtrA. Chloramphenicol chase experiments (*Figure 5D and E*) revealed that CtrA is indeed more stable in Δ*citA* cells than in *WT* cells, with the stability levels being similar to that of a non-degradable version of CtrA (*ctrA:: ctrA-M2*) (*Domian et al., 1997*).

To correlate these indirect reporter assays directly and specifically with increased transcription at CtrA-dependent promoters on a genome-wide scale, we quantified the occupancy of RNA polymerase (RNAP) on the genome using chromatin-immunoprecipitation followed by deep-sequencing (ChIP-Seq) experiments (*Figure 5F and G*, *Figure 5—source data 1*). This quantification revealed an increase of RNAP occupancy at many, but not all, CtrA-dependent promoters in cells lacking CitA when compared with *WT* cells. Interestingly, the promoters with the highest change in abundance of RNAP in *citA* cells when compared with *WT* cells are those that are activated by CtrA in $G_1$-phase (*Figure 5F*, *Figure 5—figure supplement 1D*; *Delaby et al., 2019*; *Fumeaux et al., 2014*; *Schrader et al., 2016*). This is exemplified by traces of RNAP occupancy in *WT* and *citA* mutant cells on well characterized $G_1$-promoters of *sciP*, *pilA*, and *hfsJ* (*Figure 5G*).

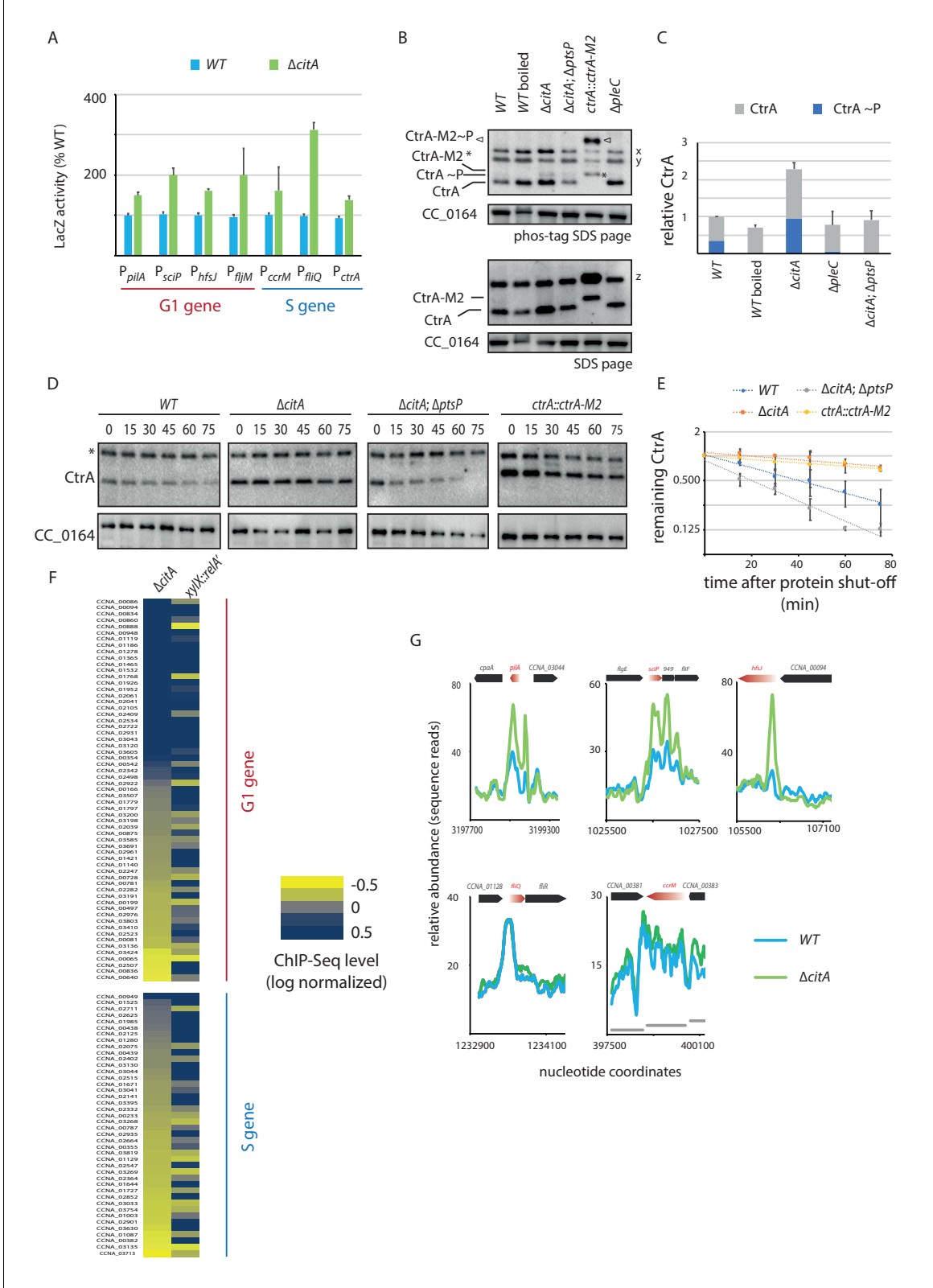

**Figure 5.** Absence of *citA* induces hyperactivation of CtrA. (**A**) Promoter-probe assays of G$_1$ (P$_{pilA}$, P$_{sciP}$, P$_{hfsJ}$, P$_{fljM}$) and S (P$_{ccrM}$, P$_{fliQ}$, P$_{ctrA}$) transcriptional reporters that are directly activated by CtrA in *WT* or Δ*citA* cells. Values are expressed as relative values compared to the *WT* (activity in *WT* set at 100%). Error bars represent the standard deviation from the mean of three independent replicates. (**B**) Phos-tag SDS-PAGE (top part) immunoblots show phosphorylation of CtrA (CtrA ~P) in extracts from *WT* (MB1), Δ*citA* (MB2559) or Δ*citA*; Δ*ptsP* (MB2426) cells. As a control, boiling of

*Figure 5 continued on next page*

*Figure 5 continued*

a *WT* lysate leads to loss of detectable CtrA ~P. As additional controls, lysates from Δ*pleC* cells in which CtrA ~P levels are reduced (***Biondi et al., 2006***; ***Radhakrishnan et al., 2010***) and from a strain harboring a tagged version (*ctrA::ctrA-M2*) as the only source CtrA (***Domian et al., 1997***) were analyzed. The same samples were analyzed by standard SDS-PAGE immunoblotting (bottom part) to measure total CtrA protein levels. In both cases, polyclonal antibodies to CtrA were used to reveal the immunoblot. The position of non-phosphorylated CtrA-M2 and phosphorylated CtrA-M2 (CtrA-M2 ~P) are indicated by a star and triangle, respectively. Non-specific bands that resulted from cross-reactivity of the antiserum to CtrA are indicated as x, y and z on the right of the immunoblots. (C) Graph showing quantification of band intensities from panel (B); the averages from two independent replicates are represented with error bars showing the standard deviations. (D) Immunoblot showing the stability of CtrA in *WT* (MB1), Δ*citA* (MB2529); Δ*ptsP citA*::Tn (MB2426) cells and in a strain expressing a stable variant of CtrA (NA1000; *ctrA::ctrA-M2*). Exponentially growing cultures were treated with chloramphenicol (50 µg.mL$^{-1}$) to shut off protein synthesis. The abundance of CtrA and CC_0164 (as a loading control) was monitored over time. An asterisk indicates a contaminant band that cross-reacts with the CtrA antibody. (E) Graph showing quantification of band intensities from panel (C); averages from three independent replicates are represented with standard deviations shown as error bars. (F) Heat map to compare ChIP-Seq (chromatin immunoprecipitation coupled to deep-sequencing) performed with antibodies recognizing the RNAP on chromatin from *WT* (MB1), Δ*citA* (CC2529) and a strain expressing *relA'-FLAG* from the *xylX* promoter (MB3282). Direct targets of CtrA, classified into two classes—G$_1$ and S— are represented. The color key indicates the degree to which the occupancy of RNAP is altered in the different genetic background compared to *WT* expressed as log$_2$ ratio. (G) ChIP-Seq traces of RNAP on different CtrA target promoters in *WT* (MB1) (blue line) or Δ*citA* (MB2529) (green line) cells. Genes encoded are represented as boxes on the upper part of the graph, red genes indicate the gene of interest represented. The online version of this article includes the following source data and figure supplement(s) for figure 5:

**Source data 1.** ChIP-Seq data set showing RNAP peak abundance measured as sequencing reads of a 20-bp window across the genome of *WT*, Δ*citA* and *xylX*::P$_{xyl}$-*relA'-FLAG* cells (in sheet 1). Sheet two shows the peaks sorted for CtrA-activated promoters that fire in G$_1$-phase, and sheet three shows the peaks for CtrA-activated promoters that fire in late S-phase.
**Figure supplement 1.** Delayed polar development of cells lacking CitA.

To see whether this effect is specific to the *citA* mutant phenotype or a generalized response of a cellular G$_1$ block, we used a control strain harboring a *relA'-FLAG* under the control of a promoter that is inducible by xylose. When induced, the resulting cells exhibit a G$_1$ arrest similar the Δ*citA* strain . Tracking RNAP occupancy by ChIP-Seq revealed an increase of binding on CtrA-regulated promoters, but without preference for the G$_1$-phase and S-phase promoter classes (***Figure 5F***). It is also important to note that global analysis of RNAP binding on all promoters in pairwise comparison between *relA'-FLAG* cells and *WT* cells or Δ*citA* cells (with a R$^2$ of 0.83 and 0.81, respectively) showed poor correlation (***Figure 5F***; ***Figure 5—figure supplement 1D***). Thus, although RNAP occupancy under ectopic (p)ppGpp production leads to an enrichment on both classes of CtrA-dependent promoters, the changes induced by the *citA* mutation, at least on the level of RNAP occupancy, are restricted to G$_1$-phase promoters. Interestingly, the LacZ promoter probe assays suggest that the *citA* mutation might also promote a transcriptional event after RNAP promoter recruitment.

## CitA and (p)ppGpp act antagonistically on CtrA

Immunoblotting experiments, using polyclonal antibodies to CitA, revealed that CitA is present at a constant level throughout the cell cycle (***Figure 4—figure supplement 1F***). This is consistent with RNA-Seq and ribosome profiling data showing that the levels of the citrate synthase (*citA*, *citB* and *citC*) transcripts and their association with ribosomes does not vary considerably during the cell cycle and that the *citA* transcript is more abundant than those of *citB* and *citC* (***Schrader et al., 2016***). As the cell cycle control function of CitA is not explained by changes in CitA abundance, other cell cycle signals or events probably affect CitA action. As the delay in the G$_1$→S transition of *citA* mutant cells probably confers a reduced growth rate of the population (***Figure 6—figure supplement 1A***), we anticipated that the isolation of fast-growing suppressor mutants would reveal how the cell cycle defect of *citA* cells can be overcome. We therefore isolated spontaneous suppressor mutants by serially diluting Δ*citA* or *citA*::Tn cultures. After three days of serial dilution, we plated cells on PYE and isolated large colonies from the background of slower-growing *citA*::Tn or Δ*citA* colonies. The growth and morphology of these mutant cells is like that of *WT* cells (***Figure 6—figure supplement 1A***). Whole-genome sequencing of two *citA*::Tn and one Δ*citA* suppressor mutant revealed a different frameshift mutation in the same domain of the PEP-phosphotransferase protein encoded by the *ptsP* gene (*CCNA_00892*) (***Ronneau et al., 2016***; ***Sanselicio et al., 2015***). PtsP resembles the first enzyme of a nitrogen-related PEP-phosphotransferase (PTS) protein homolog (EI$^{Ntr}$ in

Enterobacteria) and was shown to inhibit the hydrolase activity of SpoT, the bifunctional synthase/hydrolase of the (p)ppGpp alarmone (*Ronneau et al., 2016*).

We hypothesized that the PtsP frameshift mutation in the *citA* suppressor mutants eliminates or decreases PtsP function by affecting (p)ppGpp levels. Indeed, when the *citA*::Tn mutation was introduced into Δ*ptsP* or Δ*spoT* cells, the resulting double mutants grow faster in PYE broth than the Δ*citA* single mutant and have a higher EOP (*Figure 6—figure supplement 1B*). Importantly, the FACS profile of Δ*ptsP citA*::Tn or Δ*spoT citA*::Tn double mutant cells mirrors that of *WT* cells, indicating that loss of (p)ppGpp production indeed mitigates the effects caused by loss of CitA (*Figure 6A*), including the enhanced levels of CtrA ~P that are restored to WT levels in *citA*; *ptsP* double-mutant cells (*Figure 5B, C, D and E*). Quantification of radiolabeled (p)ppGpp extracted from *WT* and *citA* mutant cells grown in PYE did not reveal an increase in (p)ppGpp levels (*Figure 6B*), suggesting that inactivation of *citA* and the (p)ppGpp pathways converge on the same

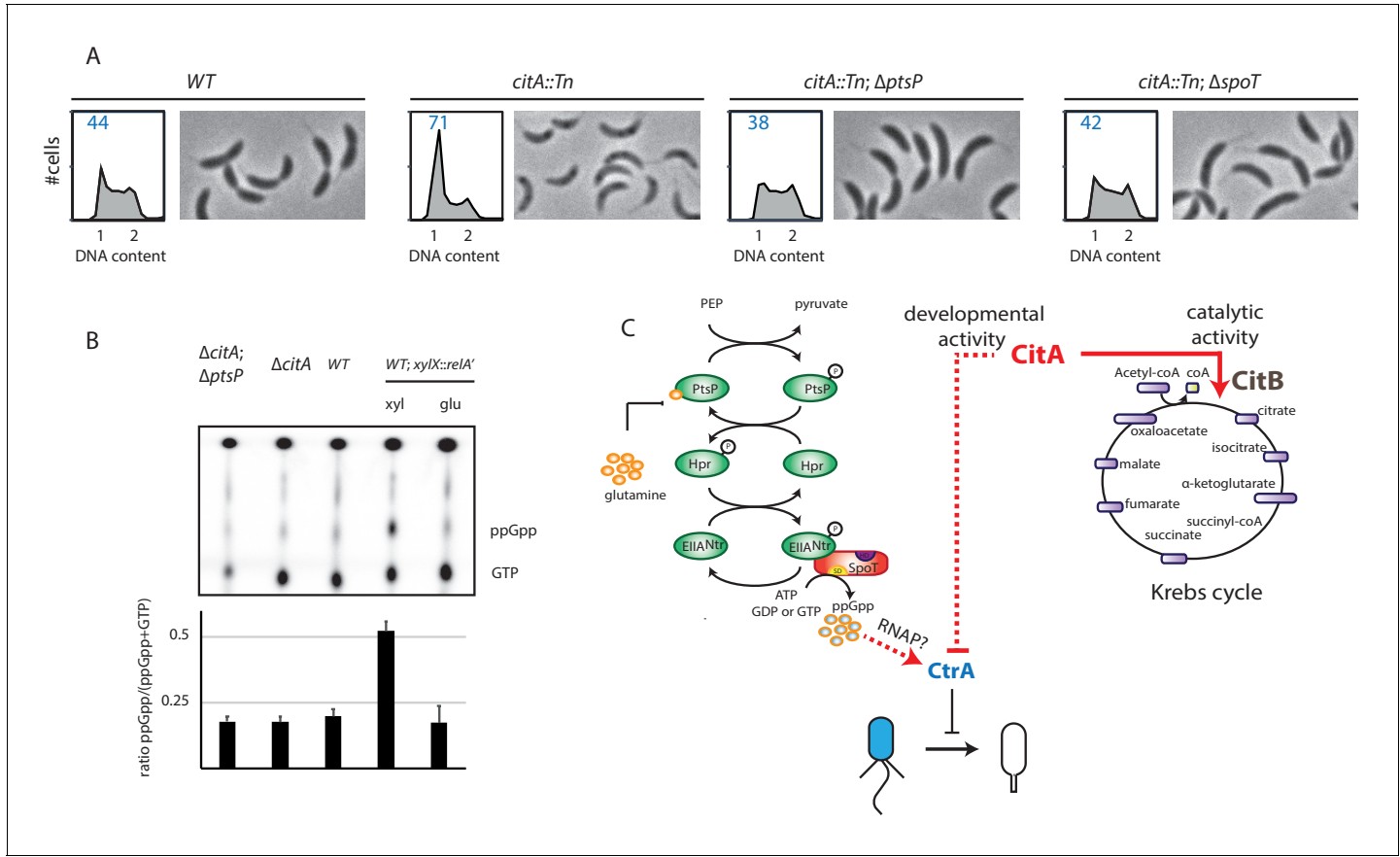

**Figure 6.** Absence of (p)ppGpp counteracts the Δ*citA* phenotype. (A) Flow cytometry profiles and phase contrast images of *WT* (MB1), *citA*::Tn (MB2622), Δ*spoT citA*::Tn (MB2413) and Δ*ptsP citA*::Tn (MB2426) cells. Genome content was analyzed by FACS during the exponential growth phase in PYE. (B) Intracellular levels of (p)ppGpp in *WT*, Δ*citA* (MB2529), Δ*citA*; *ptsP* (MB2426) and (as a positive control) RelA'-FLAG-expressing cells (MB3282). Cells were cultivated in PYE. MB3282 was cultivated in PYE for 3 hours and, then, cultures were divided in two. Glucose 0.2% or xylose 0.3% was added to repress or induce the induction of *relA'-FLAG* for one hour. The TLC autoradiograph image shown in the upper part of the figure was used to calculate the ppGpp/(GTP+ppGpp) shown in the lower panel. Error bars represent the standard deviations of the means from three independent replicates. (C) Scheme of the Pts[Ntr] signalling pathway (*Ronneau et al., 2016*) Intracellular glutamine regulates the autophosphorylation of PtsP. Under nitrogen starvation, the glutamine pool drops, triggering PtsP phosphorylation that leads to an increase of phosphorylated EII[Ntr]. Once phosphorylated, EII[Ntr] inhibits the hydrolase activity of SpoT, leading to the accumulation of (p)ppGpp, which acts as a positive regulator of CtrA and is bound by RNA polymerase (RNAP). The two functions of CitA are represented, one as a metabolic enzyme in the Krebs cycle and the other in the development of *C. crescentus* involving negative activity on CtrA that is independent of its catalytic activity. Dashed lines indicate that the suspected action on CtrA is indirect.

The online version of this article includes the following figure supplement(s) for figure 6:

**Figure supplement 1.** Bypass of the *citA* mutation by inactivation of *ptsP* or *spoT*.

target. In addition, artificial (p)ppGpp production by expressing RelA'-FLAG enhances swarming motility in soft (0.3%) agar (*Figure 5—figure supplement 1B*), but the *citA* mutant does not show a commensurate increase, further reinforcing the notion that (p)ppGpp levels are not elevated in *citA* mutant cells when compared to *WT* cells.

We conclude that CitA acts negatively on CtrA ~P and that this action depends on the presence of (p)ppGpp.

## Discussion

### Convergence of positive regulation by (p)ppGpp and negative regulation by CitA

Ectopic induction of (p)ppGpp in *WT* cells delays the $G_1 \rightarrow S$ transition and prevents the concomitant turnover of CtrA. The augmentation of the $G_1$ population when *citA* is inactivated and the stabilization of CtrA turnover perfectly mirror these effects. However, (p)ppGpp levels are not altered in the *citA* mutant, yet (p)ppGpp is absolutely required for the manifestation of the *citA* phenotype. On the basis of these results, we propose that CitA and the (p)ppGpp pathways converge to cause an increase in active CtrA (*Figure 6C*). Although we cannot exclude the hypothesis that CitA acts downstream of (p)ppGpp, a well-known transcriptional target of (p)ppGpp is RNAP (*Hauryliuk et al., 2015*). RNAP also seems to be a target of (p)ppGpp in alpha-proteobacteria (*Delaby et al., 2019*; *Wells and Long, 2003*), suggesting that the effect of (p)ppGpp on CtrA is mediated by a change in transcription. An active site in the cavity created by the alpha, beta' and omega subunits of RNAP binds (p)ppGpp (*Hauryliuk et al., 2015*). The observed accumulation of CtrA ~P in *citA* mutant cells and the increase in occupancy of RNAP at CtrA target promoters could result in effects that are comparable to the ectopic induction of (p)ppGpp.

As control of CtrA stability is mediated by the proteolytic adaptors CpdR, RcdA and PopA (*Joshi and Chien, 2016*), it is tempting to speculate that the *citA* mutation influences this pathway. However, the fact that the *citA*::Tn mutation was isolated as a suppressor from a Δ*tipN* Δ*cpdR* background already argues against this possibility. Moreover, we observed that the activities of the CtrA-dependent transcriptional reporters $P_{pilA}$-*lacZ* and $P_{hfsJ}$-*lacZ* are diminished in Δ*popA* and Δ*cpdR* mutant cells relative to those in *WT* cells (*Figure 5—figure supplement 1E*), whereas reporter activities in the *citA* mutant exceed *WT* levels. Thus, if the effects on CtrA in *citA* cells are mediated via CpdR/RcdA/PopA, then another pathway must also be affected to explain the observed effects on CtrA ~P.

### CitA as a cell cycle checkpoint

By affecting CtrA, arguably the master regulator of the *C. crescentus* cell cycle, CitA is perfectly positioned to integrate central energy metabolism with cell cycle transcriptional regulation. This function is unique to CitA, as expression of the paralog CitB from *C. crescentus* or the GltA ortholog from *E. coli* in Δ*citA* cells does not restore cell cycle control, even though both enzymes exhibit efficient citrate synthase activity in an *E. coli* reporter system. Our finding that addition of glutamine does not rescue the developmental problem of a Δ*citA* strain and that metabolite extractions from *citA* mutant cells grown on PYE do not reveal a major perturbance in the levels of tricarboxylic acids, provide further support for the conclusion that the *citA* mutant phenotype is not simply caused by a metabolic deficiency of blocked citrate production. Complementation analysis with catalytically inactive CitA variants revealed that they still confer cell cycle control functions. Moreover, other TCA cycle enzymes are essential for viability in *C. crescentus* (*Christen et al., 2011*) *Figure 3*, presumably because TCA products are essential during growth on PYE. Thus, the apparent redundancy in citrate synthase activities may have allowed the appropriation of CitA to control the cell cycle, as a checkpoint regulator and 'moonlighting' or 'trigger' enzyme.

Other bacterial lineages also encode multiple citrate synthases, and even within the Rhizobia, symbiotic relatives of *C. crescentus*, genera with three *citA* paralogs are often found (*Figure 3—figure supplement 1C*). The requirement of citrate synthase in virulence and development has been noted in other bacteria. Recently, the GltA citrate synthase from *Klebsiella pneumoniae* was identified as a virulence determinant (*Vornhagen et al., 2019*) that is required for replication in different organs, despite the presence of a GltA paralog in the genome. In addition, TCA cycle proteins have

been implicated in conferring persister (non-growing) traits in *Staphylococcus aureus* cells that protect them against bacteriocidal antibiotics (*Zalis et al., 2019*). This finding aligns well with our observation that the non-replicative $G_1$-phase population is increased in *C. crescentus citA* mutants.

Developmental roles have also been described for citrate synthase in other bacteria. *B. subtilis* cells that lack citrate synthase sporulate poorly (*Ireton et al., 1995*) and a citrate synthase mutant of *Streptomyces coelicolor* is impaired in aerial mycelium formation (*Viollier et al., 2001*). Importantly, while the growth defect of the citrate synthase mutant in *S. coelicolor* on minimal medium was suppressed by the addition of glutamate, development remains perturbed. Thus, developmental events in bacteria may be controlled by switches, and central metabolic enzymes serve as ideal checkpoint mechanisms that couple developmental gene expression to central energy metabolism.

Eukaryotic cells, such as those of *Saccharomyces cerevisiae*, restrict citrate synthase paralogs to different compartments of the cell. CIT1 is located in the mitochondria where it participates in the TCA cycle, while CIT2 is located in the peroxisome, where it acts in the glyoxylate cycle (*Kim et al., 1986*; *Rosenkrantz et al., 1986*). In *Podospora anserina*, a citrate synthase mutant strain exhibits a developmental phenotype that impairs meiosis independently of its catalytic citrate synthase activity (*Ruprich-Robert et al., 2002*), reminiscent of our finding highlighting alternate functions of citrate synthases in cell cycle control of another domain of life.

## Metabolic fluctuations during the *C. crescentus* cell cycle

The regulatory role of moonlighting enzymes raises the question of whether metabolic changes occur during the bacterial cell cycle to direct developmental changes, perhaps using moonlighting enzymes as sensors. Evidence has been provided that the cellular redox potential changes as a function of the *C. crescentus* cell cycle (*Narayanan et al., 2015*), and a recent study revealed that glutathione and many unknown metabolites fluctuate in accordance with the cell cycle (*Hartl et al., 2020*). The oxidoreductase homolog KidO is an NADH-binding protein that is present in the $G_1$-phase and during cell constriction. KidO is bifunctional, acting as cell division inhibitor that binds FtsZ and as a negative regulator of the CtrA activation pathway (*Radhakrishnan et al., 2010*). Interestingly, another division regulator that functions as a moonlighting enzyme and that is degraded in a ClpXP and CpdR-dependent manner has been identified: the glutamate dehydrogenase GdhZ whose activity is modulated by glutamate and NADH (*Beaufay et al., 2015*). The fact that KidO and GdhZ, two NAD(H) binding proteins, and CitA, an active citrate synthase, execute cell cycle control functions, indicates that *C. crescentus*, and probably other bacteria, integrate cell cycle control with central cellular metabolism (at multiple levels) using different checkpoint regulators derived from metabolic enzymes.

The moonlighting functions of KidO, GdhZ and CitA are not apparent by analysis of their primary structure. Although GdhZ was discovered as an interaction partner of FtsZ in a yeast two-hybrid screen (*Beaufay et al., 2015*), the genes encoding KidO and CitA both emerged from a forward genetic selection for cell cycle mutants that restore activity of the CtrA-regulated promoter $P_{pilA}$ that fires in $G_1$-phase (*Radhakrishnan et al., 2010*). As CtrA targeted promoters include not only those controlling expression of envelope and developmental functions, but also those controlling metabolic functions (*Fiebig et al., 2014*; *Fumeaux et al., 2014*; *Laub et al., 2000*), it is appealing to speculate that the molecular and genetic interplay between metabolism and cell cycle control is much more profound than anticipated, at least in *C. crescentus* and related bacteria.

## Materials and methods

**Key resources table**

| Reagent type (species) or resource | Designation | Source or reference | Identifiers | Additional information |
|---|---|---|---|---|
| Strain, strain background (*Caulobacter crescentus* NA1000) | *Caulobacter crescentus* NA1000 | Evinger and Agabian; PMID:334726 | | See *Supplementary file 1* |

*Continued on next page*

*Continued*

| Reagent type (species) or resource | Designation | Source or reference | Identifiers | Additional information |
|---|---|---|---|---|
| Antibody | CtrA Rrabbit polyclonal | *Delaby et al., 2019* PMID:31598724 | CtrA antibody are home-made raised against the full-length protein of *C. crescentus.* | Immunoblot: 1/5000 dilution. *Figure 4 —figure supplement 1* |
| Antibody | CitA Rrabbit polyclonal | This study | CitA antibody are home-made raised against the full-length protein of *C. crescentus.* | Immunoblot: 1/5000 dilution  *Figure 4—figure supplement 1* |
| Antibody | *E. coli* RNA Polymerase Antibody Sampler Kit Mouse monoclonal | Biolegend | 699907 | Mix 1:1:1:1 between all sera. ChIP-Seq: 1/500 dilution, *Figure 5* |
| Commercial assay or kit | Phos-tag | FUJIFILM Wako Chemicals | Distributor 300–93523 Manufacturer AAL-107M | 25 µM final, *Figure 5* |
| Chemical compound | H332PO4 | Hartmann Analytic | Cat n° P-RB-1 | |
| Chemical compound | Acetonitrile OPTIMA LC/MS grade | Fisher Scientific | A955-212 | |
| Chemical compound | Methanol OPTIMA LC/MS grade | Fisher Scientific | A456-212 | |
| Chemical compound | Water LC/MS grade | Fisher Scientific | W/0112/17 | |
| Chemical compound | Formic acid | Biosolve | 069141 | |
| Chemical compound | Ammonium hydroxide solution 25% | Sigma-Aldrich | 30501 | |
| Chemical compound | Mass Spectrometry Metabolite Library | Sigma-Aldrich | MSMLS-1EA | |
| Chemical compound | Major Mix IMS TOF calibration kit | Waters | 186008113 | |
| Chemical compound | Leucine Enkephalin | Waters | 700004768–1 | Waters TOF G2-S Sample Kit −2 (700008892) |
| Software, algorithm | UNIFI V.1.9.3 | Waters | | |
| Software, algorithm | Progenesis QI v2.3 | Nonlinear Dynamics, Waters | | |
| Software, algorithm | SIMCA-P 15.0 | Umetrics | | |
| Software, algorithm | MicrobeJ | Ducret, 2017 PMID:27572972 | | |
| Software, algorithm | SeqMonk | Babraham Bioinformatics Institute | V1.40.0 | |
| Other | Polyethyleneimine (PEI) plate | Sigma-Aldrich | Ref Z122882-25EA | *Figure 6* |
| Other | Merck SeQuant ZIC-pHILIC column (150 × 2.1 mm, 5 µm) | Merck Millipore | 1504600001 | |
| Other | Merck SeQuant ZIC-pHILIC Guard kit (20 × 2.1 mm, 5 µm) | Merck Millipore | 1504380001 | |

## Strains and growth conditions

Strains, plasmids and oligos are listed in *Supplementary files 1*, *2* and *3*. *C. crescentus* NA1000 (*Marks et al., 2010*) and derivatives were cultivated at 30°C in peptone yeast extract (PYE)-rich medium (2 g/L bactopeptone, 1 g/L yeast extract, 1 mM $MgSO_4$, and 0.5 mM $CaCl_2$) or in M2

minimal salts supplemented with 0.2% glucose (M2G, 0.87 g/L $Na_2HPO_4$, 0.54 g/L $KH_2PO_4$, 0.50 g/L $NH_4Cl$, 0.2% [wt/vol] glucose, 0.5 mM $MgSO_4$, 0.5 mM $CaCl_2$, and 0.01 mM $FeSO_4$) (*Ely, 1991*). *E. coli* S17-1 $\lambda pir$ (*Simon et al., 1983*) and EC100D (Epicentre Technologies, Madison, WI) cells were grown at 37°C in Lysogeny Broth (LB)–rich medium (10 g/L NaCl, 5 g/L yeast extract, and 10 g/L tryptone). When appropriate, media were supplemented with antibiotics at the following concentrations (µg/mL in liquid/solid medium for *C. crescentus* strains; $µg.mL^{-1}$ in liquid/solid medium for *E. coli* strains): kanamycin (5/20 $µg.mL^{-1}$; 20/20 $µg.mL^{-1}$), tetracycline (1/1 $µg.mL^{-1}$; not appropriate), spectinomycin and streptomycin 5 $µg.mL^{-1}$ (in solid medium for *C. crescentus* only) (25/25 $µg.mL^{-1}$; 30/90 $µg.mL^{-1}$), gentamycin (1/1; 10/25 $µg.mL^{-1}$), aztreonam (in solid medium only) (2.5 $µg.mL^{-1}$) and colistin (4 $µg.mL^{-1}$). PYE plates containing 3% sucrose were used to select for loss of pNTPS138-derived plasmids by recombination when constructing mutants by double recombination. When needed, for *C. crescentus*, D-xylose was added at 0.3% final concentration, glucose at 0.2% final concentration. Glutamine was used at 9.3 mM final in liquid and solid medium.

Swarmer cell isolation, electroporations, biparental matings (intergeneric conjugations) and bacteriophage ϕCr30-mediated generalized transductions were performed as described previously (*Ely and Johnson, 1977*) with slight modifications. Briefly, swarmer cells were isolated by Percoll density-gradient centrifugation at 4°C, followed by three washes and final re-suspension in pre-warmed (30°C) PYE. Electroporation was done from 1 mL overnight culture that had been washed three times in sterile water. Biparental mattings were done using exponential phase *E. coli* S17-1 donor cells and *C. crescentus* recipient cells washed in PYE and mixed at 1:3 ratio on a PYE plate. After 4–5 hr of incubation at 30°C, the mixture of cells was plated on PYE harboring aztreonam (to counter select *E. coli*) and the antibiotic that the conjugated plasmid confers resistance to. Generalized transductions using ϕCr30 were carried out by mixing 50 µL ultraviolet-inactivated ϕCr30 lysate with 500 µL stationary phase recipient cells, incubating for 2 hours, followed by plating on PYE-containing antibiotic to select for the transduced DNA.

## Metabolite extraction

For metabolite extraction, *C. crescentus* cells were grown overnight at 30°C in PYE medium and diluted to reach an $OD_{600nm}$ ~ 0.4. Ten mL of cell culture were centrifuged at 2000 g for 5 minutes at 4°C. Metabolism was then quenched by resuspending the pellet in 1 mL of precooled methanol/$H_2O$ (80:20 [vol/vol], kept at ~ −20°C). Cells were subjected to lysis by five thaw/freeze (40°C /−80°C) cycles. Cellular debris was removed by centrifugation at 17,000 g for 20 minutes at 4°C. Metabolite extracts were kept at −80°C prior to analysis on LC-MS. Bacterial biomass of individual samples was determined for normalization. The supernatants were completely evaporated using a SpeedVac (ThermoFisher, Langenselbold, Germany) and metabolite extracts were reconstituted in 100 µL acetonitrile:$H_2O$ 50:50. Quality control (QC) and diluted QC (dQC, diluted by 50%) samples were prepared by pooling equivalent volumes of all reconstituted samples and injected at a regular interval of five samples to assess analytical variability.

## Liquid chromatography–high resolution mass spectrometry (LC-HRMS) analysis

LC experiments were performed on a Waters H-Class Acquity UPLC system composed of a quaternary pump, an auto-sampler including a 15 µL flow-through-needle injector and a two-way column manager (Waters, Milford, USA) for which temperatures were set at 7°C and 40°C, respectively. The injected volume was 10 µL. Samples were analyzed with a hydrophilic liquid interaction chromatography (HILIC) SeQuant Zic-pHILIC column (150 × 2.1 mm, 5 µm) and the appropriate guard kit. For mobile phases, solvent A was acetonitrile and solvent B was $H_2O$ containing 2.8 mM ammonium formate adjusted to pH 9. Column flow rate was set at 300 $µL.min^{-1}$. The following gradient was applied: 5% B for one minute, increased to 51% B over 9 minutes, holding for 3 minutes at 51% B and then returning back to 5% B in 0.1 min and re-equilibrating the column for 6.9 min. The UPLC system was coupled to a TWIMS-QTOF high resolution HRMS (Vion, Waters, Manchester, UK) through an electrospray ionization (ESI) interface. Analyses were performed in negative ESI mode, and continuum data in the range of 50–1000 m/z were acquired with a scan time of 0.2 s. The ESI parameters were set as follows: capillary voltage was −2.0 kV, source and desolvation temperatures were set at 120°C and 500°C, respectively, cone and desolvation gas flow were 50 and 800 L/

h, respectively. Velocity and height of StepWave1 and StepWave2 were set to 300 m/s and 5 V and to 200 m/s and 30 V, respectively. The high definition MS$^E$ (HDMS$^E$, using ion mobility) settings consisted of trap wave velocity at 100 m/s; trap pulse height A at 10 V; trap pulse height B at 5 V; IMS wave velocity at 250 m/s; IMS pulse height at 45 V; wave delay set at 20 pushes; and gate delay at 0 m/s. Gas flows of ion mobility instrument were set to 1.60 L/minute for trap gas, and 25 mL/min for IMS gas. Buffer gas was nitrogen. Fragmentation was performed in HDMS$^E$ mode. For the collision energy, 6.0 eV was used for low energy and high energy was a ramp from 10 to 60 eV. Nitrogen was used as collision gas. Leucine-encephalin served as a lock-mass (554.2615 m/z for ESI-) infused at 5 minute intervals. The CCS and mass calibration of the instrument were done with the calibration mix 'Major mix IMS-TOF calibration' (Waters, Manchester, UK). UNIFI v1.9.3 was used for data acquisition and data treatment.

## Analysis of raw LC-MS data
Run alignment, peak picking, adduct deconvolution and feature annotation were sequentially performed on Progenesis QI v2.3 (Nonlinear Dynamics, Waters, Newcastle upon Tyne, UK). Detected peaks were annotated with regard to a set of pure reference standards (MSMLS Library of Standards, Sigma-Aldrich) measured under the experimental conditions described previously (*Pezzatti et al., 2019b*). The following tolerances were used: 2.5 ppm for precursor and fragment mass, 10% for retention time (Rt), and 5% in the case of collisional cross section (CCS). Data processing was achieved by SUPreMe, which is in-house software with capabilities for drift correction, noise filtering and sample normalization. Finally, data were transferred to SIMCA-P 15.0 software (Umetrics, Umea, Sweden) for multi-variate analysis (MVA).

## Microscopy and image analysis
Exponential phase *C. crescentus* cells cultivated in PYE were immobilized on a thin layer of 1.2% agarose. For *C. crescentus* time-lapse imaging, cells were first synchronized by Percoll density-gradient centrifugation and then immobilized on a thin layer of 1.2% agarose in PYE. Fluorescence and contrast microscopy images were taken with a phase contrast objective (Zeiss, alpha plan achromatic 100X/1.46 oil phase 3) on an Axio Imager M2 microscope (Zeiss), with appropriate filter (Visitron Systems GmbH) and a cooled CCD camera (Photometrics, CoolSNAP HQ2) controlled through Metamorph (Molecular Devices). Images were acquired and processed with ImageJ via Fiji software (*Schindelin et al., 2012*; *Schneider et al., 2012*). To perform cell segmentation and tracking, images were processed using MicrobeJ (*Ducret et al., 2016*). Statistics were performed on experiments performed in triplicate representing more than 300 cells.

## ChIP-SEQ
Mid-log phase cells were cross-linked in 10 mM sodium phosphate (pH 7.6) and 1% formaldehyde at room temperature (RT) for 10 minutes and on ice for 30 minutes thereafter, washed three times in phosphate-buffered saline (PBS) and lysed in a Ready-Lyse lysozyme solution (Epicentre Technologies) according to the manufacturer's instructions. Lysates were sonicated in an ice-water bath (15 cycles of 30 seconds ON, 30 seconds OFF) to shear DNA fragments to an average length of 0.3–0.5 kbp and cleared by centrifugation at 14,000 rpm for 2 minutes at 4°C. Lysates were normalized by protein content, diluted to 1 mL using ChIP buffer (0.01% SDS, 1.1% Triton X-100, 1.2 mM EDTA, 16.7 mM Tris-HCl [pH 8.1]), 167 mM NaCl plus protease inhibitors (Roche, Switzerland) and precleared with 80 µl of protein-A agarose (Roche) and 100 µg BSA. To immunoprecipitate the chromatin, 2 µL of a mixed of RNA polymerase antibody sampler kit (ratio 1:1:1:1, Biolegend) were added to the supernatant, incubated overnight at 4°C with 80 µL of protein-A agarose beads pre-saturated with BSA. The immunoprecipitate was washed once with low salt buffer (0.1% SDS, 1% Triton X-100, 2 mM EDTA, 20 mM Tris-HCl [pH 8.1] and 150 mM NaCl), high salt buffer (0.1% SDS, 1% Triton X-100, 2 mM EDTA, 20 mM Tris-HCl [pH 8.1] and 500 mM NaCl) and LiCl buffer (0.25 M LiCl, 1% NP-40, 1% sodium deoxycholate, 1 mM EDTA and 10 mM Tris-HCl [pH 8.1]), and twice with TE buffer (10 mM Tris-HCl [pH 8.1] and 1 mM EDTA). The protein–DNA complexes were eluted in 500 µL freshly prepared elution buffer (1% SDS and 0.1 M NaHCO$_3$), supplemented with NaCl to a final concentration of 300 mM and incubated overnight at 65°C to reverse the crosslinks. The samples were treated with 2 µg of Proteinase K for 2 hours at 45°C in 40 mM EDTA and 40 mM Tris-HCl (pH

6.5). DNA was extracted using phenol:chloroform:isoamyl alcohol (25:24:1), ethanol precipitated using 20 μg of glycogen as carrier and resuspended in 100 μL of water.

Immunoprecipitated chromatin was used to prepare barcoded libraries for deep-sequencing at Fasteris SA (Geneva, Switzerland). ChIP-Seq libraries were prepared using the DNA Sample Prep Kit (Illumina) following the manufacturer's instructions. Single-end runs were performed on an Illumina Genome Analyzer IIx or HiSeq2000, 50 cycles were read and yielded several million reads. The single-end sequence reads stored in FastQ files were mapped against the genome of Caulobacter crescentusNA1000 (NC_011916) and converted to SAM using BWA and SAM tools from the galaxy server (https://usegalaxy.org/). The resulting SAM was imported into SeqMonk (http://www.bioinformatics.babraham.ac.uk/projects/seqmonk/, version 0.21.0) to build sequence read profiles. The initial quantification of the sequencing data was done in SeqMonk: the genome was subdivided into 20-bp probes, and for every probe, we calculated a value that represents the number of reads that occur within the probe (using the Read Count Quantitation option). The heatmaps represent the abundance of RNA polymerase 200 bp upstream and 200 bp after the beginning of the CDS of the gene that belongs to the CtrA regulon, as determined in previous studies (*Fumeaux et al., 2014*; *Schrader et al., 2016*). Sequence data have been deposited in the Gene Expression Omnibus (GEO) database (GSE144533).

## Detection of (p)ppGpp

(p)ppGpp levels were determined using a protocol adapted from a previous study by *Lesley and Shapiro (2008)*. Briefly, strains were grown in PYE medium, and 1 mL normalized at $OD_{600}$ of 0.4 was centrifuged for 3 minutes at 10,000 RPM and resuspended in 250 μl of PYE. A final concentration of 20 μCi of $H_3^{32}PO_4$ (Hartmann Analytic) was added to the cultures, and the cells were incubated for 120 minutes at 30°C with agitation, before being fixed by the addition of 2M formic acid. As a positive control, a culture of NA1000 expressing *relA'-FLAG* under the control of the *xylX* promoter was grown in PYE for 3 hours, then xylose or glucose was added to induce or repress, respectively, the expression of *relA'-FLAG* for 120 minutes. All cell extracts were kept on ice for 30 min and then centrifuged for 5 minutes, and $8 \times 2$ μl of the supernatant (16 μl total) was spotted onto a polyethyleneimine (PEI) plate (Sigma-Aldrich). The PEI plate had been soaked in sterile distilled water overnight and dried at room temperature before spotting. The plate was developed in 1.5 M $KH_2PO_4$ (pH 3.4) in a saturated thin-layer chromatography (TLC) chamber for approximately 180 min and dried at RT. Nucleotides were detected via a phosphorimaging system (Tritium screen). ppGpp, and GTP were identified on the basis of their retardation factor ($R_f$) and by comparison with a standard for GTP. Spots were quantified using ImageJ software.

## Phos-tag polyacrylamide gel electrophoresis (PAGE)

To determine the in vivo phosphorylation of CtrA, strains were grown to mid-log phase ($OD_{600nm}$ around 0.4), and 1 mL of cells were pelleted at 20,000 g at 4°C for 5 minutes. Pellets were resuspended in 75 μL TE buffer (10 mM Tris-HCl [pH 8.0] and 1 mM EDTA) followed by the addition of 75 μL loading buffer 2X (0.25 M Tris [pH 6.8], 6% [wt/vol] SDS, 10 mM EDTA, 20% [vol/vol] glycerol) containing 10% (vol/vol) β-mercaptoethanol. Samples were normalized for equivalent loading using $OD_{600nm}$. Some samples were boiled by heating at 90°C for 10 minutes. Samples were stored on ice for a short time (<10 minutes) prior to loading onto Phos-tag acrylamide gels.

Phos-tag SDS-PAGE gels were prepared with 25 μM Phos-tag acrylamide and 50 μM $MnCl_2$. All gels were run at 4°C under constant voltage (80 V). Before transfer by blotting, gels were washed three times for 10 minutes in transfer buffer containing 10 mM EDTA at 4°C to remove $Mn^{2+}$ from the gel and once with transfer buffer without EDTA at 4°C. Blots were revealed by immunodetection using Western Blot Signal Enhancer (Thermo Pierce) after incubation with rabbit anti-CtrA (1:5000) primary antibodies and a polyclonal donkey anti-rabbit HRP conjugated secondary antibody (Jackson ImmunoResearch). Band intensities were analyzed using ImageJ. The total CtrA content was determined using a control gel that did not contain Phos-tag, whereas phosphorylated and non-phosphorylated forms of CtrA were estimated from the gel containing Phos-tag using two independent biological replicates.

## CtrA stability measurements by chloramphenicol chase

To measure protein stability in vivo, cells were grown to mid-log phase ($OD_{600nm}$ of ca. 0.4). Protein synthesis was blocked by the addition of 50 μg/mL chloramphenicol. Samples were taken every 15 min and frozen immediately at −80℃ before being analyzed by immunoblotting.

## β-galactosidase assay

100 μL of cells at $OD_{600nm}$ = 0.1–0.4 were lysed with chloroform and mixed with 700 μl of Z buffer (60 mM $Na_2HPO_4$, 40 mM $NaH_2PO_4$, 10 mM KCl and 1 mM $MgSO_4$ heptahydrate). 200 μL of ONPG (4 mg $ml^{-1}$ o-nitrophenyl-β-D-galactopyranoside in 0.1 M $KPO_4$[pH 7.0]) were added and the was reaction timed. When a medium-yellow color developed, the reaction was stopped with 500 μL of 1 M $Na_2CO_3$. The $OD_{420nm}$ of the supernatant was determined and the units were calculated with the equation: $U = (OD_{420nm} \times 1000)/(OD_{660nm} \times$ time [in min] $\times$ volume of culture [in mL]). The assays were done in triplicate and normalization was performed by conversion of the Miller Units (absolute values) of one arbitrarily chosen *WT* construct or *WT* background as reference, set to 100%. All absolute values were then converted to relative values, averaged and the error was determined by calculation of the standard deviation (s.d.). Data are from three biological replicates.

## Genome-wide transposon mutagenesis coupled to deep-sequencing (Tn-Seq)

Pools of >100,000 Tn mutants were isolated as kanamycin-aztreonam or kanamycin-colistin resistant clones in the NA1000 (WT), Δ*tipN*, Δ*cpdR*::Ω backgrounds, using the previously described protocol involving a mini-*himar1* Tn encoding kanamycin resistance (*Viollier et al., 2004*). For each Tn pool, chromosomal DNA was extracted and used to generate a Tn-Seq library sequenced on an Illumina HiSeq 2500 sequencer (Fasteris, Geneva, Switzerland). The single-end sequence reads (50 bp) stored in FastQ files were mapped against the genome of the *Caulobacter crescentus* NA1000 (NC_011916) (*Marks et al., 2010*) genome and converted to BED files using BWA-MEM and bedtools BAM to BED tools, respectively, from the Galaxy server (https://usegalaxy.org/). The resulting BED file was imported into SeqMonk (http://www.bioinformatics.babraham.ac.uk/projects/seqmonk/ ) to build sequence read profiles. The initial quantification of the sequencing data was done in Seq-Monk: the genome was subdivided into 50-bp probes, and for every probe, we calculated a value that represents a normalized read number per million. A ratio of the reads obtained in the Δ*tipN* or Δ*cpdR* strains to the *WT* reads was calculated for each 50-bp position. This file was used to generate the zoomed panels of the *popA*, *rcdA* and *cpdR* loci (*Figure 1B*) or the *tipN* locus (*Figure 1—figure supplement 1A and B*).

## Identification of *citA* (P*pilA*-*nptII* suppressor screen)

The *citA*::Tn insertion was identified using a modification of the kanamycin resistance suppressor screen (*Radhakrishnan et al., 2010*). Briefly, we screened for mini-*himar1* Tn insertions that restore P*pilA* firing to Δ*tipN*; Δ*cpdR* double mutant cells harboring the P*pilA*-*nptII* transcriptional reporter, which confers kanamycin resistance to 20 μg $mL^{-1}$ when P*pilA* is fully active. The Tn encodes gentamycin resistance on plasmid pMar2xT7 delivered from *E. coli* S17-1 λ*pir* (*Liberati et al., 2006*) to Δ*tipN*; Δ*cpdR*; *pilA*::P*pilA*-*nptII C. crescentus* cells by selection on plates with gentamycin (1 μg $mL^{-1}$), kanamycin (20 μg $mL^{-1}$) and aztreonam (2.5 μg $mL^{-1}$, to counter-select *E. coli*). This screen gave rise to one isolate Φ40 with the desired resistance profile. The Tn insertion in Φ40 was mapped to the uncharacterized *CCNA_01983* gene at nucleotide (nt) position 1061847 of the *C. crescentus* NA1000 genome sequence using arbitrarily primed PCR (*Liberati et al., 2006*).

## Evolution experiment selecting for fast-growing *citA* suppressor mutants

Two independent clones of *C. crescentus* NA1000 freshly transduced with Δ*citA::kan* or *citA*::Tn were inoculated in 3 mL of PYE. Stationary phase cultures were diluted in 3 mL PYE to $OD_{600nm}$ ~0.02. After 2 days, the four cultures were re-diluted to $OD_{600nm}$ ~0.001 in 3 mL PYE. The phenotype of each strain was checked by phase-contrast microscopy and FACS analysis. Each culture was streaked on a PYE plate and one single colony from each culture was grown overnight and chromosomal DNA was extracted. Three suppressors were subjected to whole-genome sequencing.

Library preparation and sequencing were performed by the Genomic platform iGE3 at the university of Geneva on a HiSeq 2500 with 50-bp paired-end reads. Data analysis to identify mutations was done using freebayes v1.1.0–3 (*Garrison and Marth, 2012*) against the *C. crescentus* NA1000 reference genome (NC_011916.1).

## Growth curves

The overnight cultures were started in PYE or in M2G. The cultures were diluted to obtain an $OD_{600nm}$ of 0.1 in PYE or M2G and were incubated at 30°C with continuous shaking in a microplate reader (Synergy H1, Biotek). The $OD_{600nm}$ was recorded every 30 min for 30 hours. The graph represents the trend of the growth curve of three independent experiments.

## Fluorescence-activated cell sorting (FACS)

Cells in exponential growth phase ($OD_{600}$0.3 to 0.6) were fixed 1:10 (vol/vol) in ice-cold 70% ethanol solution and stored at −20°C until further use. For rifampicin treatment, the mid-log phase cells were grown in the presence of 20 µg/mL rifampicin at 30°C for 3 hours. Cells were fixed as mentioned above. Fixed cells were centrifuged at 6200 g for 3 minutes at room temperature and washed once in FACS staining buffer (10 mM Tris-HCl, 1 mM EDTA, 50 mM Na-citrate, 0.01% Triton X-100 [pH 7.2]). Then, cells were centrifuged at 6200 g for 3 minutes at RT, and resuspended in FACS staining buffer containing RNase A (Roche) at 0.1 mg.mL$^{-1}$ for 30 minutes at RT. Cells were stained in FACS staining buffer containing 0.5 µM of SYTOX green nucleic acid stain solution (Invitrogen) and then analyzed using a BD Accuri C6 flow cytometer instrument (BD Biosciences, San Jose, CA, United States). Flow cytometry data were acquired and analyzed using the CFlow Plus v1.0.264.15 software (Accuri Cytometers Inc). A total of 20,000 cells were analyzed from each biological sample, performed in triplicates. The green fluorescence (FL1-A) parameters was used to determine cell chromosome contents. Flow cytometry profiles within one figure were recorded in the same experiment, on the same day with the same settings. The scales of the y- and x-axes of the histograms within one figure panel are identical. Each experiment was repeated independently three times and representative results are shown. The relative chromosome number was directly estimated from the FL1-A value of NA1000 cells treated with 20 µg/mL rifampicin for 3 hours at 30°C. Rifampicin treatment of cells blocks the initiation of chromosomal replication but allows ongoing rounds of replication to finish.

## Preparation of cell-free extracts

500 µL of an exponential *Caulobacter* or *E. coli* cells ($OD_{600nm}$ = 0.4 and 0.8, respectively) were harvested with 20,000 g at 4°C for 5 minutes. Whole-cell extracts were prepared by resuspension of cell pellets in 75 µL TE buffer (10 mM Tris-HCl [pH 8.0] and 1 mM EDTA) followed by addition of 75 µL loading buffer 2X (0.25 M Tris [pH 6.8], 6% [wt/vol] SDS, 10 mM EDTA, 20% [vol/vol] glycerol) containing 10% (vol/vol) β-mercaptoethanol. Samples were normalized for equivalent loading using $OD_{600nm}$ and were heated for 10 min at 90°C prior to loading.

## Immunoblot analysis

Protein samples were separated by SDS–polyacrylamide gel electrophoresis and blotted on polyvinylidenfluoride membranes (Merck Millipore). Membranes were blocked overnight with Tris-buffered saline 1X (TBS) (50 mM Tris-HCl, 150 mM NaCl [pH 8]) containing, 0.1% Tween-20% and 8% dry milk and then incubated for an additional three hours with the primary antibodies diluted in TBS 1X, 0.1% Tween-20, 5% dry milk. The different polyclonal antisera to CitA (1:5000) and to CtrA (1:5000) were used. Primary antibodies were detected using HRP-conjugated donkey anti-rabbit antibody (Jackson ImmunoResearch) with ECL Western Blotting Detection System (GE Healthcare) and a luminescent image analyzer (Chemidoc MP, Biorad).

## CitA purification and production of antibodies

Recombinant CitA protein was expressed as an N-terminally His$_6$-tagged variant from pET28a in *E. coli* BL21(DE3)/pLysS and purified under native conditions using Ni$^{2+}$ chelate chromatography. Cells were grown in LB at 37°C to an $OD_{600nm}$ of 0.6, induced by the addition of IPTG to 1 mM for 3 hr, and harvested at 5000 RPM at 4°C for 30 minutes. Cells were pelleted and re-suspended in 25 mL of

lysis buffer (10 mM Tris HCl [pH 8], 0.1 M NaCl, 1.0 mM β-mercaptoethanol, 5% glycerol, 0.5 mM imidazole Triton X-100 0.02%). Cells were sonicated in a water–ice bath (15 cycles of 30 s ON, 30 s OFF). After centrifugation at 5000 g for 20 minutes at 4°C, the supernatant was loaded onto a column containing 5 mL of Ni-NTA agarose resin (Qiagen, Hilden, Germany) pre-equilibrated with lysis buffer. The column was rinsed with lysis buffer, 400 mM NaCl and 10 mM imidazole, both prepared in lysis buffer. Fractions were collected (in 300 mM imidazole buffer, prepared in lysis buffer) and used to immunize New Zealand white rabbits (Josman LLC).

## Strain construction
### MB3075 (NA1000 Δ*tipN*; Δ*popA*)
A pNTPS138 derivative (pNTPS138-Δ*tipN*) (*Huitema et al., 2006*) was integrated nearby the markerless Δ*tipN* mutation by homologous recombination. Phage φCr-30-mediated generalized transduction was used to transfer the mutant Δ*tipN* allele into the recipients NA1000 Δ*popA* by selecting for kanamycin resistance. Clones that have lost pNPTS138-Δ*tipN* by homologous recombination were probed for kanamycin resistance (on PYE plates supplemented with kanamycin) following sucrose counter-selection. PCR was used to verify the integrity of the mutants.

### MB3079 (NA1000 Δ*tipN*; Δ*rcdA*::Ω)
A pNTPS138 derivative (pNTPS138-Δ*tipN*) (*Huitema et al., 2006*) was integrated nearby the markerless Δ*tipN* mutation by homologous recombination. Phage φCr-30-mediated generalized transduction was used to transfer the mutant Δ*tipN* allele into the recipients NA1000 Δ*rcdA*::Ω by selecting for kanamycin resistance. Clones that have lost pNPTS138-Δ*tipN* by homologous recombination were probed for kanamycin resistance (on PYE plates supplemented with kanamycin) following sucrose counter-selection. PCR was used to verify the integrity of the mutants.

### MB2017 (NA1000 Δ*tipN*; Δ*cpdR*::*tet*)
The Δ*cpdR::tet* allele was introduced into NA1000 Δ*tipN* by generalized transduction using φCr30 and then selected on PYE plates containing tetracycline.

### MB2366 (NA1000 Δ*tipN*; *xylX*::*kidO*$^{AA::DD}$)
The *xylX*::*kidO*$^{AA::DD}$ (kan$^R$) allele was introduced into NA1000 Δ*tipN* by generalized transduction using φCr30 and then selected on PYE plates containing kanamycin.

### MB2720 (NA1000 Δ*tipN*; Δ*cpdR*::*tet*; Δ*kidO*)
A pNTPS138 derivative (pNTPS138-Δ*tipN*) (*Huitema et al., 2006*) was integrated nearby the markerless Δ*tipN* mutation by homologous recombination. φCr-30-mediated generalized transduction was used to transfer the mutant Δ*tipN* allele into the recipients NA1000 Δ*kidO* by selecting for kanamycin resistance. Clones that have lost pNPTS138-Δ*tipN* by homologous recombination were probed for kanamycin resistance (on PYE plates supplemented with kanamycin) following sucrose counter-selection. PCR was used to verify the integrity of the mutants. Then, Δ*cpdR::tet* allele was introduced into NA1000 Δ*tipN*; Δ*kidO* by transduction using φCr30 and then selected on PYE plates containing tetracycline.

### MB2325 (NA1000 *pilA*::P$_{pilA}$-*GFP*)
The *pilA*::P$_{pilA}$-*GFP* (kan$^R$) allele was introduced into NA1000 by generalized transduction using φCr30 and then selected on PYE plates containing kanamycin.

### MB2327 (NA1000 Δ*cpdR*::Ω; *pilA*::P$_{pilA}$-*GFP*)
The *pilA*::P$_{pilA}$-*GFP* (kan$^R$) allele was introduced into NA1000 Δ*cpdR*::Ω (Spc$^R$) by generalized transduction using φCr30 and then selected on PYE plates containing kanamycin.

### MB2329 (NA1000 Δ*tipN*; *pilA*::P$_{pilA}$-*GFP*)
The *pilA*::P$_{pilA}$-*GFP* (kan$^R$) allele was introduced into NA1000 Δ*tipN* by generalized transduction using φCr30 and then selected on PYE plates containing kanamycin.

### MB2331 (NA1000 $\Delta tipN$; $\Delta cpdR$::$\Omega$; $pilA$::$P_{pilA}$-GFP)

The $pilA$::$P_{pilA}$-GFP (kan$^R$) allele was introduced into MB2017 (NA1000 $\Delta tipN$; $\Delta cpdR$::$\Omega$ by generalized transduction using $\phi$Cr30 and then plated on PYE-containing kanamycin.

### MB2268 (NA1000 $pilA$::$P_{pilA}$-nptII)

The $pilA$::$P_{pilA}$-nptII (Spc$^R$) allele was introduced into NA1000 by generalized transduction using $\phi$Cr30 and then selected on PYE plates containing spectinomycin.

### MB2271 (NA1000 $\Delta tipN$; $\Delta cpdR$::$tet$; $pilA$::$P_{pilA}$-nptII)

The $pilA$::$P_{pilA}$-nptII (Spc$^R$) allele was introduced into MB2017 (NA1000 $\Delta tipN$; $\Delta cpdR$::$tet$) by generalized transduction using $\phi$Cr30 and then selected on PYE plates containing spectinomycin.

### MB2559 (NA1000 $\Delta citA$::pNTPS138-$\Delta citA$)

A pNTPS138 derivative (pNTPS138-$\Delta citA$) was integrated nearby the marker-less $\Delta citA$ mutation by homologous recombination. $\phi$Cr-30-mediated generalized transduction was used to transfer the mutant $\Delta citA$ allele into the recipients NA1000 by selecting for kanamycin resistance on PYE plates containing kanamycin.

### MB3056 (NA1000 $\Delta tipN$; $\Delta cpdR$::$tet$ $citA$::Tn; $pilA$::$P_{pilA}$-nptII)

The $citA$::Tn (Gent$^R$) allele was introduced into MB2271 (NA1000 $\Delta tipN$; $\Delta cpdR$::$tet$; $pilA$::$P_{pilA}$-nptII) cells by transduction using $\phi$Cr30 and then selected on PYE plates containing gentamycin.

### MB3058 (NA1000 $\Delta tipN$; $\Delta cpdR$::$tet$; $\Delta citA$ $pilA$::$P_{pilA}$-nptII)

$\phi$Cr-30-mediated generalized transduction was used to transfer the mutant $\Delta citA$ allele from MB2559 into MB2017 (NA1000; $\Delta tipN$; $\Delta cpdR$::$tet$) recipient cells by selecting for kanamycin resistance. Clones that have lost pNPTS138-$\Delta citA$ by homologous recombination were probed for kanamycin resistance (on PYE plates supplemented with kanamycin) following sucrose counter-selection (giving rise to strain named MB3054). PCR was used to verify the integrity of the mutants. Then, the $pilA$::$P_{pilA}$-nptII (Spc$^R$) allele was introduced into MB3054 (NA1000 $\Delta tipN$; $\Delta cpdR$::$tet$ $\Delta citA$) by generalized transduction using $\phi$Cr30, selecting on PYE plates containing spectinomycin.

### MB2679 (NA1000 $\Delta citBC$)

The markerless $\Delta citBC$ double mutant was created by introducing into the WT (NA1000) using the standard two-step recombination sucrose counter-selection procedure induced by the pNTPS138-$\Delta citBC$ (pMB309). Briefly, first integration was done by mating of the eMB552 (S17-1 carrying the pMB309) and C. crescentus NA1000, selecting for kanamycin and aztreonam (to eliminate the donor strain). Clones that have lost pNPTS138-$\Delta tipN$ by homologous recombination were probed for kanamycin resistance (on PYE plates supplemented with kanamycin) following sucrose counter-selection (giving rise to a strain named MB2679). PCR, using outside primers that do not hybridize within the $\Delta citBC$ deletion carried on pNTPS138, was used to verify the integrity of the mutants.

### MB2622 (NA1000 $citA$::Tn)

The $citA$::Tn (Gent$^R$) allele was introduced into NA1000 by generalized transduction using $\phi$Cr30 and then selected on PYE plates containing gentamycin.

### MB1537 (NA1000; pMT335)

Plasmid pMT335 was introduced into NA1000 by electroporation and then plated on PYE harboring gentamycin.

### MB3433 (NA1000 $\Delta citA$; pMT335)

$\phi$Cr-30-mediated generalized transduction was used to transfer the mutant $\Delta citA$ allele from MB2559 into MB1537 recipient cells by selecting for kanamycin resistance.

### MB3435 (NA1000 ΔcitA; pMT335-citA)

Plasmid pMB302 (pMT335-citA) was introduced into NA1000 by electroporation and then plated on PYE harboring gentamycin. φCr-30-mediated generalized transduction was used to transfer the mutant ΔcitA allele from MB2559 into NA1000; pMT335-citA cells by selecting for kanamycin resistance.

### MB3469 (NA1000 ΔcitA; pMT335-citB)

Plasmid pMB303 (pMT335-citB) was introduced into NA1000 by electroporation and then plated on PYE harboring gentamycin. φCr-30-mediated generalized transduction was used to transfer the mutant ΔcitA allele from MB2559 into NA1000; pMT335-citB cells by selecting for kanamycin resistance.

### MB3471 (NA1000 ΔcitA; pMT335-citC)

Plasmid pMB304 (pMT335-citC) was introduced into NA1000 by electroporation and then plated on PYE harboring gentamycin. φCr-30-mediated generalized transduction was used to transfer the mutant ΔcitA allele from MB2559 into NA1000; pMT335-citC cells by selecting for kanamycin resistance.

### MB3473 (NA1000 ΔcitA; pMT335-gltA)

Plasmid pMB310 (pMT335-gltA) was introduced into NA1000 by electroporation and then plated on PYE harboring gentamycin. φCr-30-mediated generalized transduction was used to transfer the mutant ΔcitA allele from MB2559 into NA1000; pMT335-gltA cells by selecting for kanamycin resistance.

### MB3437 (NA1000 ΔcitA; pMT335-citA$^{H303W}$)

Plasmid pMB325 (pMT335-citA$^{H303W}$) was introduced into NA1000 by electroporation and then plated on PYE harboring gentamycin. φCr-30-mediated generalized transduction was used to transfer the mutant ΔcitA allele from MB2559 into NA1000; pMT335-citA$^{H303W}$ cells by selecting for kanamycin resistance.

### MB3439 (NA1000 ΔcitA; pMT335-citA$^{H303A}$)

Plasmid pMB326 (pMT335-citA$^{H303A}$) was introduced into NA1000 by electroporation and then plated on PYE harboring gentamycin. φCr-30-mediated generalized transduction was used to transfer the mutant ΔcitA allele from MB2559 into NA1000; pMT335-citA$^{H303A}$ cells by selecting for kanamycin resistance.

### MB2452 (NA1000 parB::GFP-parB; citA::Tn)

The citA::Tn (Gent$^R$) allele was introduced into MB557 (NA1000; parB::GFP-parB) by generalized transduction using φCr30 and then plated on PYE plates containing gentamycin.

### MB3467 (NA1000 parB::GFP-parB; ΔcitA)

φCr-30-mediated generalized transduction was used to transfer the mutant ΔcitA allele from MB2559 into MB557 (NA1000; parB::GFP-parB) by selecting for kanamycin resistance on plates containing kanamycin.

### MB2413 (NA1000 ΔspoT; citA::Tn)

φCr-30-mediated generalized transduction was used to transfer the citA::Tn allele into MB2403 (NA1000 ΔspoT) cells by selection on PYE plates containing gentamycin.

### MB2426 (NA1000 ΔptsP; citA::Tn)

φCr-30-mediated generalized transduction was used to transfer the citA::Tn allele into MB2417 (NA1000 ΔptsP) cells by selection on plates PYE containing gentamycin.

### UG430 (NA1000 *stpX*::*stpX-GFP*; *spmX*::*spmX-mCherry*)

φCr-30-mediated generalized transduction was used to transfer the *stpX*::*stpX-GFP* construct into *spmX*::*spmX-mCherry* cells by selection on PYE plates containing kanamycin.

### MB3598 (NA1000 *citA*::Tn; *stpX*::*stpX-GFP*; *spmX*::*spmX-mCherry*)

φCr-30-mediated generalized transduction was used to transfer the *citA*::Tn allele into UG430 (NA1000 *stpX*::*stpX-GFP*; *spmX*::*spmX-mCherry*) cells by selection on PYE plates containing gentamycin.

### MB3566 (NA1000 *spmX*::*spmX-mCherry*; *tipF*::*tipF-GFP*)

φCr-30-mediated generalized transduction was used to transfer the *tipF*::*tipF-GFP* (KanR) allele into MB656 (NA1000 *spmX*::*spmX-mCherry*) cells by selection on PYE plates containing kanamycin.

### MB3613 (NA1000 *citA*::Tn; *spmX*::*spmX-mCherry*; *tipF*::*tipF-GFP*)

φCr-30-mediated generalized transduction was used to transfer the *citA*::Tn allele into MB3566 (NA1000 *tipF*::*tipF-GFP*; *spmX*::*spmX-mCherry*) cells by selection on PYE plates containing gentamycin.

### MB3568 (NA1000 *spmX*::*spmX-mCherry*; *xylX*::*mipZ-YFP*)

φCr-30-mediated generalized transduction was used to transfer the *xylX*::*mipZ-YFP* (KanR) allele into MB656 (NA1000 *spmX*::*spmX-mCherry*) cells by selection on PYE plates containing kanamycin.

### MB3615 (NA1000 *citA*::Tn; *spmX*::*spmX-mCherry*; *xylX*::*mipZ-YFP*)

φCr-30-mediated generalized transduction was used to transfer the *citA*::Tn allele into MB3568 (NA1000 *xylX*::*mipZ-YFP*; *spmX*::*spmX-mCherry*) cells by selection on PYE plates containing gentamycin.

### MB3623 (NA1000 Δ*citA*; plac290-$P_{pilA}$)

φCr-30-mediated generalized transduction was used to transfer the mutant Δ*citA* allele from MB2559 into NA1000 plac290-$P_{pilA}$ (*Skerker and Shapiro, 2000*) by selecting for kanamycin resistance on plates containing kanamycin.

### MB3625 (NA1000 Δ*citA*; plac290-$P_{fljM}$)

φCr-30-mediated generalized transduction was used to transfer the mutant Δ*citA* allele from MB2559 into NA1000 plac290-$P_{fljM}$ (*Fumeaux et al., 2014*) by selecting for kanamycin resistance on plates containing kanamycin.

### MB3627 (NA1000 Δ*citA*; plac290-$P_{ctrA}$)

φCr-30-mediated generalized transduction was used to transfer the mutant Δ*citA* allele from MB2559 into NA1000 plac290-$P_{ctrA}$ (*Fumeaux et al., 2014*) by selecting for kanamycin resistance on plates containing kanamycin.

### MB3590 (NA1000 Δ*citA*; plac290-$P_{sciP}$)

φCr-30-mediated generalized transduction was used to transfer the mutant Δ*citA* allele from MB2559 into NA1000 plac290-$P_{sciP}$ (*Fumeaux et al., 2014*) by selecting for kanamycin resistance on plates containing kanamycin.

### MB3592 (NA1000 Δ*citA*; plac290-$P_{hfsJ}$)

φCr-30-mediated generalized transduction was used to transfer the mutant Δ*citA* allele from MB2559 into NA1000 plac290-$P_{hfsJ}$ (*Fumeaux et al., 2014*) by selecting for kanamycin resistance on plates containing kanamycin.

### MB3594 (NA1000 Δ*citA*; plac290-$P_{ccrM}$)

φCr-30-mediated generalized transduction was used to transfer the mutant Δ*citA* allele from MB2559 into NA1000 plac290-$P_{ccrM}$ (*Stephens et al., 1995*) by selecting for kanamycin resistance on plates containing kanamycin.

### MB3596 (NA1000 Δ*citA*; plac290-$P_{fliQ}$)

φCr-30-mediated generalized transduction was used to transfer the mutant Δ*citA* allele from MB2559 into NA1000 plac290-$P_{fliQ}$ (*Fumeaux et al., 2014*) by selecting for kanamycin resistance on plates containing kanamycin.

### MB3601 (NA1000 Δ*popA*; plac290-$P_{pilA}$)

Plasmid plac290-$P_{pilA}$ (*Skerker and Shapiro, 2000*) was introduced into MB46 (NA1000 Δ*popA*) by electroporation and then plated on PYE harboring tetracycline.

### MB3605 (NA1000 Δ*popA*; plac290-$P_{hfsJ}$)

Plasmid plac290-$P_{hfsJ}$ (*Fumeaux et al., 2014*) was introduced into MB46 (NA1000 Δ*popA*) by electroporation and then plated on PYE harboring tetracycline.

### MB3607 (NA1000 Δ*cpdR*::Ω; plac290-$P_{pilA}$)

Plasmid plac290-$P_{pilA}$ (*Skerker and Shapiro, 2000*) was introduced into MB47 (NA1000 Δ*cpdR*::Ω [Spc$^R$]) by electroporation and then plated on PYE harboring tetracycline.

### MB3611 (NA1000 Δ*cpdR*::Ω; plac290-$P_{hfsJ}$)

Plasmid plac290-$P_{hfsJ}$ (*Fumeaux et al., 2014*) was introduced into MB47 (NA1000 Δ*cpdR*::Ω [Spc$^R$]) by electroporation and then plated on PYE harboring tetracycline.

### eMB554 (BW35113; pMT335)

Plasmid pMT335 was introduced into BW35113 by electroporation and then plated on LB agar containing gentamycin to isolate eMB556 (BW35113; Δ*gltA*::770; pMT335).

Plasmid pMT335 was introduced into JW0710-1 (BW35113; Δ*gltA770*::*kan*) by electroporation and then plated on LB agar containing gentamycin.

### eMB558 (BW35113; Δ*gltA*::770; pMT335-*citA*)

Plasmid pMB302 (pMT335-*citA*) was introduced into JW0710-1 (BW35113; Δ*gltA770*::*kan*) by electroporation and then plated on LB agar containing gentamycin.

### eMB560 (BW35113; Δ*gltA770*::*kan*; pMT335-*citB*)

Plasmid pMB303 (pMT335-*citB*) was introduced into JW0710-1 (BW35113; Δ*gltA770*::*kan*) by electroporation and then plated on LB agar containing gentamycin to isolate eMB562 (BW35113; Δ*gltA770*::*kan*; pMT335-*citC*).

Plasmid pMB304 (pMT335-*citC*) was introduced into JW0710-1 (BW35113; Δ*gltA770*::*kan*) by electroporation and then plated on LB agar containing gentamycin.

### eMB564 (BW35113; Δ*gltA*::770; pMT335-*gltA*)

Plasmid pMB310 (pMT335-*gltA*) was introduced into JW0710-1 (BW35113; Δ*gltA770*::*kan*) by electroporation and then plated on LB agar containing gentamycin.

### eMB581 (BW35113; Δ*gltA770*::*kan*; pMT335-*citA*$^{H303W}$)

Plasmid pMB325 (pMT335-*citA*$^{H303W}$) was introduced into JW0710-1 (BW35113; Δ*gltA770*::*kan*) by electroporation and then plated on LB agar containing gentamycin.

### eMB581 (BW35113; Δ*gltA770*::*kan*; pMT335-*citA*$^{D361E}$)

Plasmid pMB327 (pMT335-*citA*$^{D361E}$) was introduced into JW0710-1 (BW35113; Δ*gltA770*::*kan*) by electroporation and then plated on LB agar containing gentamycin.

### Plasmid constructions pMB278 (pNTPS138-ΔcitA)

The plasmid construct used to delete citA (CCNA_01983) was made by PCR amplification of two fragments: the first to amplify the upstream region of citA, a 617-bp fragment was amplified using primers OMB173 and OMB174, flanked by a HindIII and a PstI site; and the second to amplify the downstream region of citA, a 567-bp fragment was amplified using primers OMB175 and OMB176, flanked by a PstI site and an EcoRI site. These two fragments were first digested with appropriate restriction enzymes and then triple ligated into pNTPS138 (M.R.K. Alley, Imperial College London, unpublished) previously restricted with EcoRI/HindIII.

### pMB288 (pNTPS138-ΔcitB)

The plasmid construct used to delete citB (CCNA_03757) was made by PCR amplification of two fragments: the first to amplify the upstream region of citB, a 550-bp fragment was amplified using primers OMB184 and OMB185, flanked by a HindII and an NdeI; and the second to amplify the downstream region of citB, a 538-bp fragment was amplified using primers OMB186 and OMB187, flanked by a NdeI site and an EcoRI site. These two fragments were first digested with appropriate restriction enzymes and then triple ligated into pNTPS138 (M.R.K. Alley, Imperial College London, unpublished) previously restricted with EcoRI/HindIII.

### pMB289 (pNTPS138-ΔcitC)

The plasmid construct used to delete citC (CCNA_03758) was made by PCR amplification of two fragments: he first to amplify the upstream region of citC, a 568-bp fragment was amplified using primers OMB188 and OMB189, flanked by a HindII and a NdeI site; and the second to amplify the downstream region of citC, a 551-bp fragment was amplified using primers OMB190 and OMB191, flanked by a NdeI site and an EcoRI site. These two fragments were first digested with appropriate restriction enzymes and then triple ligated into pNTPS138 (M.R.K. Alley, Imperial College London, unpublished) previously restricted with EcoRI/HindIII.

### pMB309 (pNTPS138-ΔcitB/citC)

The plasmid construct used to delete citB and citC (CCNA_03757 and CCNA_03758) was made by digestion of the upstream region of citB of the pMB288, a 532-bp fragment using the NdeI and EcoRI sites. This fragment was ligated into the pMB289 digested by MfeI and NdeI enzymes.

### pMB302 (pMT335-citA)

The citA coding sequence was PCR amplified from NA1000 using the OMB179 and OMB182 primers. This fragment was digested with NdeI/EcoRI and cloned into NdeI/EcoRI-digested pMT335.

### pMB303 (pMT335-citB)

The citB coding sequence was PCR amplified from NA1000 using the OMB194 and OMB195 primers. This fragment was digested with NdeI/EcoRI and cloned into NdeI/EcoRI-digested pMT335.

### pMB304 (pMT335-citC)

The citC coding sequence was PCR amplified from NA1000 using the OMB196 and OMB197 primers. This fragment was digested with NdeI/EcoRI and cloned into NdeI/EcoRI-digested pMT335.

### pMB310 (pMT335-gltA)

The gltA coding sequence was PCR amplified from E. coli MG1655 using the OMB203 and OMB204 primers. This fragment was digested with NdeI/EcoRI and cloned into NdeI/EcoRI-digested pMT335.

### pMB287 (pSC-citA)

The citA coding sequence was PCR amplified from C. crescentus using the OMB179 and OMB183 primers. This fragment was digested with NdeI/HindIII and cloned into NdeI/HindIII digested pSC.

## pMB325 (pMT335-*citA*$^{H303W}$)

The *citA* catalytic mutant was generated using the QuickChange Site-directed Mutagenesis kit (Agilent technologies). Briefly, the plasmid pMB302 (pMT335-*citA*) was PCR amplified using the mutagenic primers OMB232 and OMB233, containing the H303W mutation. This PCR was followed by a *Dpn*I digestion to digest the parental plasmid, and this digestion was used to transform electrocompetent *E. coli*. The integration of the site-directed mutation in *citA* coding sequence was verified by sequencing.

## pMB326 (pMT335-*citA*$^{H303A}$)

The *citA* catalytic mutant was generated using QuickChange Site-directed Mutagenesis kit (Agilent technologies). Briefly, the plasmid pMB302 (pMT335-*citA*) was PCR amplified using the mutagenic primers OMB236 and OMB237, containing the H303A mutation. This PCR was followed by a *Dpn*I digestion to digest the parental plasmid, and this digestion was used to transform electrocompetent *E. coli*. The integration of the site-directed mutation in *citA* coding sequence was verified by sequencing.

# Acknowledgements

We thank Justine Collier, Sean Crosson, Martin Thanbichler, Michael Laub, Urs Jenal and Lucy Shapiro for materials, Julien Prados for help with Tn-Seq, ChIP-Seq and suppressors analyses, and Gaël Panis, Nicolas Kint for critical reading of the manuscript. We thank especially Benjamin Albert and Maksym Shyian from the David Shore lab for critical help with the phos-tag experiment. This work was supported by the Swiss National Science Foundation grant 31003A_182576 to Patrick H Viollier.

# Additional information

### Funding

| Funder | Grant reference number | Author |
|---|---|---|
| Schweizerischer Nationalfonds zur Förderung der Wissenschaftlichen Forschung | 31003A_182576 | Patrick H Viollier |

The funders had no role in study design, data collection and interpretation, or the decision to submit the work for publication.

### Author contributions

Matthieu Bergé, Serge Rudaz, Conceptualization, Data curation, Investigation; Julian Pezzatti, Conceptualization, Validation, Investigation; Víctor González-Ruiz, Conceptualization, Investigation, Methodology; Laurence Degeorges, Resources, Investigation, Methodology; Geneviève Mottet-Osman, Resources; Patrick H Viollier, Conceptualization, Data curation, Funding acquisition, Methodology

### Author ORCIDs

Matthieu Bergé (ID) https://orcid.org/0000-0002-0910-6114
Patrick H Viollier (ID) https://orcid.org/0000-0002-5249-9910

### Decision letter and Author response

Decision letter https://doi.org/10.7554/eLife.52272.sa1
Author response https://doi.org/10.7554/eLife.52272.sa2

# Additional files

### Supplementary files

- Supplementary file 1. Table of *C. crescentus* and *E. coli* strains used in this study.

- Supplementary file 2. Table of plasmids used in this study.
- Supplementary file 3. Table of oligonucleotides used in this study.
- Supplementary file 4. Key resources table: table of reagents and antibodies used in this study.
- Transparent reporting form

## Data availability

All data generated or analysed during this study are included in the manuscript and supporting files. Source data files have been provided for Tn-seq and metabolomics data.

The following dataset was generated:

| Author(s) | Year | Dataset title | Dataset URL | Database and Identifier |
|---|---|---|---|---|
| Bergè M, Degeorges L, Viollier P | 2020 | Polymerase occupancy (ChIP-Seq) in WT and mutants of Caulobacter crescentus NA1000 | https://www.ncbi.nlm.nih.gov/geo/query/acc.cgi?acc=GSE144533 | NCBI Gene Expression Omnibus, GSE144533 |

The following previously published dataset was used:

| Author(s) | Year | Dataset title | Dataset URL | Database and Identifier |
|---|---|---|---|---|
| Fumeaux C, Radhakrishnan SK, Ardissone S, Théraulaz L, Frandi A, Martins D, Nesper J, Abel S, Jenal U, Viollier PH | 2014 | Examination of 5 transcripton factor binding in two different species | https://www.ncbi.nlm.nih.gov/geo/query/acc.cgi?acc=GSE52849 | NCBI Gene Expression Omnibus, GSE52849 |

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
