## [Decision Letter]

**Acceptance summary:**

Deeper study of enzymes that play multiple, unexpected roles contributes to a more enhanced understanding of biological circuits and how these are regulated in response to changing environments and cues. Your study provides an intriguing description of how citrate synthase, an enzyme involved in primary carbon metabolism in the citric acid cycle, plays an additional role in the cell cycle as a regulator of the G1-S transition. A fascinating aspect of this finding was that this regulatory role was not related to the primary catalytic activity of the enzyme, highlighting the complexity in how bacterial cells regulate the transition between growth phases, using seemingly unrelated enzymes. The clever use of screens to uncover these effects confirms the utility of this, and similar approaches, for discovering new regulatory networks.

**Decision letter after peer review:**

Thank you for submitting your article "Bacterial cell cycle control by citrate synthase independent of enzymatic activity" for consideration by *eLife*. Your article has been reviewed by three peer reviewers, and the evaluation has been overseen by a Reviewing Editor and Anna Akhmanova as the Senior Editor. The following individuals involved in review of your submission have agreed to reveal their identity: Jared M Schrader (Reviewer #1).

The reviewers have discussed the reviews with one another and the Reviewing Editor has drafted this decision to help you prepare a revised submission.

Summary:

The manuscript by Berge et al. reports the results of a forward-genetic screen to identify novel regulators of the bacterial cell cycle using *C. crescentus* as a model system, revealing that one isoform of citrate synthase (CitA) is a regulator of the G1-S transition. This result was particularly intriguing as the phenotype did not dependent on citrate synthase activity. Indeed, a second citrase synthase homologue (CitB) could not rescue the phenotypic effects of a *citA*-deletion but was sufficient to provide complementation of enzymatic activity. The authors report that this is an example of protein "moonlighting" by providing a new, unexpected function for which there are a growing number of examples. These findings could be of broad interest, pending some further mechanistic insight.

Essential revisions:

1) In the initial Tn-Seq comparing *WT* and *tipN* deletion cells, the *tipN* deletion mutant had 33% of the hits as wildtype in the *tipN* gene-how could there be any hits in the *tipN* gene if it was a clean deletion (Supplementary file 1)? By contrast, the *cpdR* deletion had 0 hits in the *cpdR* gene. Is this an error? Please address

2) The authors should use ChIP-seq and lacZ promoters fusions to assess CtrA activity in *citA* mutants and to determine if the entire CtrA region or only a subset is under CitA control. These assays are regularly used in the Viollier lab. The in vivo phosphorylation level of CtrA could also be measured in a *∆citA* mutant.

3) In the text as well as the model drawn in Figure 5D, the authors propose that CitA inhibits the Pts-Ntr pathway ultimately leading to inhibition of ppGpp synthesis. Thus, deletion of *citA* restores normal cell division in the *tipN*/*cpdR* mutant by activating SpoT and elevating ppGpp to block S-phase entry. While this model is consistent with the data, there are alternative interpretations. For example, *citA* deletion may have no effect on ppGpp levels at all and instead inhibits S-phase entry through another mechanism; however, when ppGpp levels are decreased in the evolved-fast growing PtsP mutants identified in the genetic screen, this is sufficient to promote S-phase entry. In order to demonstrate causality, it is critical to compare ppGpp levels in wildtype, *citA*-deletion, and *citA*/PtsP-double deletion strains. If the model is correct, one might expect that the *citA*-deletion increases ppGpp while the double deletion restores ppGpp back to wildtype levels. This is important to establish the model.

4) As an addendum to point 3, the authors should also measure motility behaviour of *∆citA* mutants. Indeed, if *∆citA* cells accumulate (p)ppGpp, the motility should be increased since the G1 cells would be blocked as swarmer cells. At least, Caulobacter mutants accumulating (p)ppGpp strongly increase their motility behaviour. However, based on the pictures of *∆citA* cells shown in Figures 4 and 5, stalks are clearly visible on G1 cells. This suggests that cell cycle and development are uncoupled in *∆citA* mutants, a phenotype typically observed when CtrA activity is enhanced but not when (p)ppGpp accumulates. Verification of this would make the story stronger.

5) The manuscript would benefit from a brief bioinformatics analysis reporting whether CitA and CitB were both conserved across α-proteobacterial species containing CtrA, or whether this only occurs in Caulobacteraceae. In the Discussion the authors note that there is evidence that nutritional stress may act on CtrA in Sinorhizbium meliloti, which does contain a *citA* ortholog, suggesting this moonlighting may be conserved. Examining the conservation of CitA/B may help others studying diverse α-protoebacteria to explore whether this might be happening elsewhere.

6) The story line of this manuscript is unclear. The authors start with negative genetic interactions between a polarity factor (TipN) and ClpXP proteolytic adaptors (CpdR, RcdA and PopA) due to the stabilisation of an oxidoreductase-like (KidO), which in turn decrease CtrA activity. Then, they set up a genetic screen in which they found *citA* mutants that, based on their initial hypothesis, should increase CtrA activity. But instead of characterising this potential negative regulation of CitA on CtrA activity, the authors moved to another genetic screen and found that inactivating (p)ppGpp synthesis suppressed *∆citA* phenotypes. Finally, based only on these genetic interactions, the authors proposed a model in which CitA might regulate (p)ppGpp synthesis. There may be a plausible alternative model that would take into account all the data. Indeed, a recent publication of the Viollier lab (Delaby et al., 2019) showed that (p)ppGpp is required to support CtrA activity during stationary phase. Thus, CitA might inhibit CtrA activity so that *∆citA* cells would have an exacerbated CtrA activity that leads to a G1 block, and inactivating (p)ppGpp production with spoT mutations would decrease CtrA activity back to level close to wild-type. Alternatively, (p)ppGpp and CitA could work independently of each other to antagonistically regulate G1-S transition. Therefore, the manuscript would be improved by keeping a more straightforward story line and to reinforce the likely link between CtrA and CitA. The genetic screen with the PpilA::nptII was originally used by the authors "to find mutations that maintain CtrA active in the absence of TipN and CpdR". Please consider how to better present the story.

---

## [Author Response]

Summary:The manuscript by Berge et al. reports the results of a forward-genetic screen to identify novel regulators of the bacterial cell cycle using C. crescentus as a model system, revealing that one isoform of citrate synthase (CitA) is a regulator of the G1-S transition. This result was particularly intriguing as the phenotype did not dependent on citrate synthase activity. Indeed, a second citrase synthase homologue (CitB) could not rescue the phenotypic effects of a citA-deletion but was sufficient to provide complementation of enzymatic activity. The authors report that this is an example of protein "moonlighting" by providing a new, unexpected function for which there are a growing number of examples. These findings could be of broad interest, pending some further mechanistic insight.

We are grateful to the reviewing editor and the reviewers for the fair and constructive recommendations on how to improve clarity and depth of our manuscript.

Extensive revisions and new experimentation were introduced to the manuscript, and the three major additions are summarized as follows:

1) Although the response in the *citA* mutant could be mitigated by loss (p)ppGpp as already stated in the previous version, no increase in radiolabeled (p)ppGpp was detected (using P32 orthophosphate added to cells growing in PYE) in *citA* mutant cells relative to *WT* (Figure 6B). Thus, while the *citA* mutation seems to potentiate the effects of (p)ppGpp to induce a G1 cell enrichment in the presence of (p)ppGpp, in WT only massive (p)ppGpp production induced from RelA’ induces a similar G1 arrest and this clearly is due to (an observable) an increase of intracellular (p)ppGpp by TLC analysis.

2) Knowing that CitA does not seem to affect the PtsP/spoT pathway to affect (p)ppGpp levels, we investigated further the mechanism promoting the accumulation of G1 cells in the *citA* mutant with respect to CtrA, the essential G1-phase transcriptional regulator. We show that CtrA is stabilized in *citA* mutant cells (new Figure 5C and D), matching the stabilization of CtrA under conditions of ectopic (p)ppGpp production in *WT* cells. Of course stabilization of CtrA also occurs in mutants lacking the proteolytic adaptors CpdR, RcdA or PopA, however, in these adaptor mutants CtrA-activated G1-phase promoters do NOT fire at an elevated level as seen for the *citA* mutant (new Figure 5—figure supplement 1F). Thus, and importantly, stabilization of CtrA does NOT suffice to induce the same phenotype as the *citA* mutation, indicating that the *citA* mutation also acts through another effector that controls CtrA activity and/or phosphorylation.

3) Phos-Tag experiments revealed that the levels of phosphorylated CtrA (CtrA~P) is indeed augmented in *citA* cells *versus WT* cells (new Figure 5A and B). This augmentation explains the increase in transcription in multiple CtrA-dependent promoter probe plasmids that we have assayed in the revised version of the manuscript. To match these genetic (indirect) transcriptional assays, we also quantified RNA polymerase (RNAP) occupancy at CtrA-activated promoters by ChIP-seq and found that the promoters induced in G1-phase by CtrA strongly attract RNAP in *citA* mutant cells relative to *WT* cells, compared to other CtrA target promoters that fire in S-phase (new Figure 5E and F). This result is consistent with the fact that these G1-phase promoters are known to be more sensitive to reduction in CtrA~P (PMID: 24939058).

Lastly, we added bionformatic analyses showing that close or more distant relatives of *C. crescentus* (*Caulobacterales* and *Rhizobiales/Rhodobacterales,* respectively) frequently encode multiple CitA paralogs (Figure 3—figure supplement 1C). Thus, functional specialization of citrate synthases may not be unique to *C. crescentus.*

We hope that with these new and detailed analyses on CtrA we have sufficiently advanced our understanding on the cell cycle control by CitA to convince the readers of *eLife* that we have discovered a new and important control element linking central metabolism and cell cycle progression in bacteria.

Essential revisions:1) In the initial Tn-Seq comparing WT and tipN deletion cells, the tipN deletion mutant had 33% of the hits as wildtype in the tipN gene-how could there be any hits in the tipN gene if it was a clean deletion (Supplementary file 1)? By contrast, the cpdR deletion had 0 hits in the cpdR gene. Is this an error? Please address

The Tn-Seq data presented in Supplementary file 1 quantifies the number of Tn insertions per CDS. The in-frame deletion of *tipN* was constructing to delete amino acids 24-831(PMID: 16530048), but since *tipN* encodes 882 or 888 residues (depending on the start codon used). In fact, the Tn insertions in *tipN* occurred at the end of the gene, into the 174 bp that had not been deleted. We document this, we added a new sheet (sheet 100bp, Supplementary file 1) representing the Tn insertions in probes with a sliding window of 100bp along all the chromosome.

2) The authors should use ChIP-seq and lacZ promoters fusions to assess CtrA activity in citA mutants and to determine if the entire CtrA region or only a subset is under CitA control. These assays are regularly used in the Viollier lab. The in vivo phosphorylation level of CtrA could also be measured in a ∆citA mutant.

All the experiments proposed were done and the results are presented in Figure 5. First, the in vivo CtrA~P levels were determined by immunoblotting of Phos-tag-PAGE. This revealed a strong increase in the amount of CtrA~P in *citA* mutant cells vs *WT* (Figure 5A and B). Next, we conducted antibiotic chase experiments (using a protein synthesis inhibitor) to measure the rates of CtrA decay in *WT* and *citA* cells and this revealed that the stability of CtrA is increased in the absence of CitA (Figure 5C and D). Finally, ChIP-seq experiment tracking RNA polymerase (RNAP) occupancy on different classes of CtrA-activated promoters showed a preferential enrichment of RNAP on promoters firing in G1 phase (Figure 5E and F). LacZ-based promoter probe assays in *WT* and *citA* mutant cells (Figure 5G) show an upregulation as well and further revealed that G1 promoters are not upregulated in the ∆*popA* and ∆*cpdR* background (Figure 5—figure supplement 1F), where CtrA is stable during the cell cycle. Thus, the increase of CtrA stability in ∆*citA* cells cannot account for the increase in CtrA~P and activity in CtrA-dependent promoters. In support of this, see for example that *WT* cells expressing the stabilized CtrA-M2 also do not show an increase in CtrA~P levels relative to *WT*, having substantially less than ∆*citA* cells (Figure 5A and B).

3) In the text as well as the model drawn in Figure 5D, the authors propose that CitA inhibits the Pts-Ntr pathway ultimately leading to inhibition of ppGpp synthesis. Thus, deletion of citA restores normal cell division in the tipN/cpdR mutant by activating SpoT and elevating ppGpp to block S-phase entry. While this model is consistent with the data, there are alternative interpretations. For example, citA deletion may have no effect on ppGpp levels at all and instead inhibits S-phase entry through another mechanism; however, when ppGpp levels are decreased in the evolved-fast growing PtsP mutants identified in the genetic screen, this is sufficient to promote S-phase entry. In order to demonstrate causality, it is critical to compare ppGpp levels in wildtype, citA-deletion, and citA/PtsP-double deletion strains. If the model is correct, one might expect that the citA-deletion increases ppGpp while the double deletion restores ppGpp back to wildtype levels. This is important to establish the model.

The astute reviewers were right in their speculation: we quantified the (p)ppGpp level in WT, ∆*citA* and ∆*citA* ∆*ptsP* double mutant by Thin Layer Chromatography (TLC) analysis of extracts following orthophosphate (P32) labelling in PYE rich medium (in order to not introduce another variable as such labelling experiments are typically done in minimal M5G phosphate-depleted minimal medium for efficient incorporation of the radiolabel). No increase in (p)ppGpp levels could be detected in the ∆*citA* background compared to *WT*, whereas an increase in (p)ppGpp was clearly seen when *relA’* was artificially induced in *WT* cells.

In addition, quantification of RNAP occupancy by ChIP-seq analysis in ∆*citA* or in a strain producing (p)ppGpp ectopically from RelA’ expressing *relA’* indicates differences agreement with the fact that the absence of CitA does not lead to (p)ppGpp production. So the model in Figure 6C has been changed accordingly, showing two independent pathways, the (p)ppGpp one acting positively on CtrA while the CitA pathway acts negatively on CtrA.

4) As an addendum to point 3, the authors should also measure motility behaviour of ∆citA mutants. Indeed, if ∆citA cells accumulate (p)ppGpp, the motility should be increased since the G1 cells would be blocked as swarmer cells. At least, Caulobacter mutants accumulating (p)ppGpp strongly increase their motility behaviour. However, based on the pictures of ∆citA cells shown in Figures 4 and 5, stalks are clearly visible on G1 cells. This suggests that cell cycle and development are uncoupled in ∆citA mutants, a phenotype typically observed when CtrA activity is enhanced but not when (p)ppGpp accumulates. Verification of this would make the story stronger.

All the proposed experiments were done and the results are shown in Figure 5—figure supplement 1. We do not see an increase in motility in ∆*citA* cells as dramatic as for WT cells expressing RelA’. This experiment, like the RNAP ChIP-seq and (p)ppGpp measurements detailed above, provide phenotypic evidence that the ∆*citA* mutant is not due to elevated (p)ppGpp.

We also added fluorescence microscopy showing that many ∆*citA* cells exhibit a long stalk, at the same position where SpmX-mCherry (Figure 4F), a marker of the stalked pole, is located. The stalk in ∆*citA* cells is clearly opposite the flagellar pole as observed with the TipF-GFP flagellar marker (Figure 5—figure supplement 1A). In addition, fluorescence microscopy on synchronized cells expressing a SpmX-mCherry fusion (also a marker for the remodeling of the cell pole from a flagellar into a stalked pole during the G1S transition) and a MipZ-YFP (marker of the chromosome replication) reveal that contrary to *WT* cells, ∆*citA* cells can be found with one focus of MipZ and one focus of SpmX. This indicates indeed that polar remodeling has been uncoupled from replication as the majority of these cells have a 1N chromosome equivalent and a stalked pole, a phenotype typically observed when CtrA activity is enhanced artificially (Hung and Shapiro, 2002).

5) The manuscript would benefit from a brief bioinformatics analysis reporting whether CitA and CitB were both conserved across α-proteobacterial species containing CtrA, or whether this only occurs in Caulobacteraceae. In the Discussion the authors note that there is evidence that nutritional stress may act on CtrA in Sinorhizbium meliloti, which does contain a citA ortholog, suggesting this moonlighting may be conserved. Examining the conservation of CitA/B may help others studying diverse α-protoebacteria to explore whether this might be happening elsewhere.

The bioinformatic analysis was performed and the result is included in Figure 3—figure supplement 1C. This analysis shows that for the 37 genera represented, 14 have at least two paralogs of the citrate synthase. Importantly, this analysis suggest that paralogs presence is conserved among *Rhizobiales*, *Caulobacterales* and *Rhodobacterales* group (with 12 species on 20 that have at least two paralogs) while it is not conserved in other group such as obligate intracellular pathogen like the Rickettsiae group.

6) The story line of this manuscript is unclear. The authors start with negative genetic interactions between a polarity factor (TipN) and ClpXP proteolytic adaptors (CpdR, RcdA and PopA) due to the stabilisation of an oxidoreductase-like (KidO), which in turn decrease CtrA activity. Then, they set up a genetic screen in which they found citA mutants that, based on their initial hypothesis, should increase CtrA activity. But instead of characterising this potential negative regulation of CitA on CtrA activity, the authors moved to another genetic screen and found that inactivating (p)ppGpp synthesis suppressed ∆citA phenotypes. Finally, based only on these genetic interactions, the authors proposed a model in which CitA might regulate (p)ppGpp synthesis. There may be a plausible alternative model that would take into account all the data. Indeed, a recent publication of the Viollier lab (Delaby et al., 2019) showed that (p)ppGpp is required to support CtrA activity during stationary phase. Thus, CitA might inhibit CtrA activity so that ∆citA cells would have an exacerbated CtrA activity that leads to a G1 block, and inactivating (p)ppGpp production with spoT mutations would decrease CtrA activity back to level close to wild-type. Alternatively, (p)ppGpp and CitA could work independently of each other to antagonistically regulate G1-S transition. Therefore, the manuscript would be improved by keeping a more straightforward story line and to reinforce the likely link between CtrA and CitA. The genetic screen with the PpilA::nptII was originally used by the authors "to find mutations that maintain CtrA active in the absence of TipN and CpdR". Please consider how to better present the story.

We agree that the story line is challenging for somebody who is not familiar with Caulobacter but we are also obliged to tell the story the way the discovery was made. We do not think it makes sense to separate the *tipN/cpdR* background from the discovery of the *citA*::Tn mutant, as this mutant background in which CtrA activity is reduced (apparently due to the stabilization of KidO which acts negatively on cell division and the CtrA pathway) explains perfectly why the *citA*::Tn mutation surfaced in the first place. In the revised version we now dwell extensively on characterizing the negative regulation of CtrA by CitA (new Figure 5 and see description above).

In the revised version we have made efforts to shorten the lengthy descriptions on TipN and to quickly advance to the meat of the story from the entry point. Care had to be taken to maintain the reference to KidO for several important reasons. KidO is central to explaining the effects of the *cpdR* mutation in *tipN* cells. KidO also surfaced from P*pilA*-*nptII* suppressor screen (now discussed at the end of the manuscript) and of course it is a prime example of a moonlighting enzyme acting on the *Caulobacter* cell cycle, although its metabolic activity remains obscure.